# Adaptive Normalization for Non-stationary Time Series Forecasting: A Temporal Slice Perspective

**Zhiding Liu[1,2], Mingyue Cheng[1,2], Zhi Li[3], Zhenya Huang[1,2], Qi Liu[1,2], Yanhu Xie[4], Enhong Chen[1,2]***

[1]Anhui Province Key Laboratory of Big Data Analysis and Application,
University of Science and Technology of China
[2]State Key Laboratory of Cognitive Intelligence
[3]Shenzhen International Graduate School, Tsinghua University
[4]The First Affiliated Hospital of University of Science and Technology of China
zhiding@mail.ustc.edu.cn,{mycheng,huangzhy,qiliuql,cheneh}@ustc.edu.cn
zhilizl@sz.tsinghua.edu.cn, xyh200701@sina.cn

## Abstract

Deep learning models have progressively advanced time series forecasting due to their powerful capacity in capturing sequence dependence. Nevertheless, it is still challenging to make accurate predictions due to the existence of non-stationarity in real-world data, denoting the data distribution rapidly changes over time. To mitigate such a dilemma, several efforts have been conducted by reducing the non-stationarity with normalization operation. However, these methods typically overlook the distribution discrepancy between the input series and the horizon series, and assume that all time points within the same instance share the same statistical properties, which is too ideal and may lead to suboptimal relative improvements. To this end, we propose a novel slice-level adaptive normalization, referred to **SAN**, which is a novel scheme for empowering time series forecasting with more flexible normalization and denormalization. SAN includes two crucial designs. First, SAN tries to eliminate the non-stationarity of time series in units of a local temporal slice (i.e., sub-series) rather than a global instance. Second, SAN employs a slight network module to independently model the evolving trends of statistical properties of raw time series. Consequently, SAN could serve as a general model-agnostic plugin and better alleviate the impact of the non-stationary nature of time series data. We instantiate the proposed SAN on four widely used forecasting models and test their prediction results on benchmark datasets to evaluate its effectiveness. Also, we report some insightful findings to deeply analyze and understand our proposed SAN. We make our codes publicly available[2].

## 1 Introduction

Time series forecasting is becoming increasingly prevalent in real-world scenarios. Various applications have been facilitated by the advancement of forecasting, such as energy consumption planning [33], clinical healthcare analysis [16], financial risk assessment [15] and cloud resource allocation [2]. Recently, deep learning-based methods have largely advanced forecasting and other tasks due to their powerful capacity to capture sequence dependence [23, 27, 3, 42].

Nevertheless, it is still challenging to make accurate predictions for time series forecasting due to the rapid evolution of time series points over time (a.k.a. non-stationarity of time series) [31]. Such

---

*Enhong Chen is the corresponding author.

[2]https://github.com/icantnamemyself/SAN

37th Conference on Neural Information Processing Systems (NeurIPS 2023).

non-stationarity can lead to discrepancies between different time spans and hinder the generalization of deep learning models. To alleviate the impact of the non-stationary nature, removing these dynamic factors from the original data through normalization has been proposed as a feasible solution [28].

Recently, some pioneering efforts have been devoted to this research topic [17, 25, 10]. Although these normalization approaches have significantly improved the prediction performance, we identify two limitations in existing solutions. On the one hand, most existing methods overlook the distribution discrepancy between the input series and the horizon series, and simply adopt the statistical properties of the input series to denormalize the output results. Furthermore, previous studies assume that all time points within the same instance share the same statistical properties during the normalization processing, and a global instance normalization is widely applied. Such coarse-grained settings are not appropriate since time series points rapidly change over time [6, 20], particularly in long-term forecasting scenarios where both input and horizon series may span a considerable duration. For example, there may be sudden changes in data distribution due to events like holidays or temperature spikes for electricity consumption data.

We plot a forecasting sample in Fig. 1 to better illustrate our opinion. Though temporally related, the input series' mean differs from the horizon's significantly (from 0.75 to 1.5), indicating a potentially universal distribution discrepancy. Moreover, such a distribution shift can happen rapidly at a more fine-grained slice level, violating the basic assumption of existing normalization methods. Therefore, these approaches risk damaging instinct patterns of each slice of the input sequence by *normalizing with improper statistics* while also causing a prediction shift in final forecasting results due to *poor estimation of future statistics*.

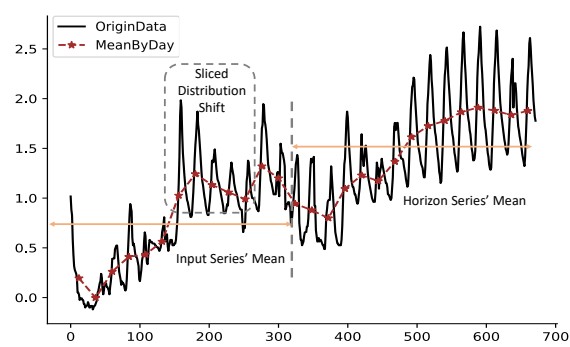

Figure 1: An illustration of a forecasting instance in energy consumption along with its daily mean (MeanByDay). We also plot the input series' mean and the horizon series' mean in the figure.

To overcome these limitations, we propose a general normalization framework for non-stationary time series forecasting named **S**licing **A**daptive **N**ormalization (SAN). SAN models the non-stationarity in the fine-grained *temporal slices*, or patches[26], which are more informative than single data points and can be regarded as fundamental units of the time series data[4, 14]. To be specific, the input sequence is first split into non-overlap equally-sized slices, which are then normalized according to their statistics and fed into the forecasting model. Meanwhile, we use a statistics prediction module to predict the distributions of future slices based on the statistics of the input. Finally, the non-stationary information is restored to the output of the forecasting model with well-estimated statistics. By modeling the slice-level characteristic, SAN is able to remove the non-stationarity in a local region. Besides, with the statistics prediction module independently modeling the evolving trends of statistical properties, SAN adopts more precise statistics for adaptive denormalization. Consequently, the non-stationary forecasting task is actually simplified by being split into statistic prediction and stationary forecasting. Moreover, SAN is a model-agnostic framework and can be applied to arbitrary forecasting models. Sufficient experiments have been conducted in a widely used benchmark dataset. The results demonstrate that SAN surpasses advanced normalization approaches by boosting the performance of various kinds of mainstream forecasting models by a large margin.

In summary, our main contributions are as follows:

- We propose SAN, a general normalization framework for non-stationary time series forecasting tasks which distinguishes by modeling the non-stationary nature from a temporal slice perspective. In this way, SAN can better remove the non-stationary factors in input sequences while keeping their distinct patterns.

- We design a flexible statistics prediction module for SAN which independently models the evolving trends of statistical properties. By explicitly learning to estimate future distributions, SAN can simplify the non-stationary forecasting task through divide and conquer.

- We conduct sufficient experiments on nine real-world datasets. Results show that SAN can be applied to various mainstream forecasting models and boost the performance by a large margin. Moreover, the comparison between SAN and state-of-the-art normalization methods demonstrates the superiority of our proposed framework.

## 2  Related Works

### 2.1  Time Series Forecasting

Time series forecasting has been extensively studied in recent decades. Originally, ARIMA [1, 40] builds an auto-regressive model and forecasts in a moving average fashion. Though theoretical guarantees are achieved, such traditional methods usually require data with ideal properties, which is inconsistent with the real-world scenario. With the increasing data availability and computing power, numerous deep-learning-based models have emerged, which always follow a sequence-to-sequence paradigm. Recurrent neural networks (RNNs) are first utilized to capture the temporal dependence by summarizing the past information in time series [30, 32, 37]. Such architectures naturally suffer from a limited reception field and an error accumulation issue caused by the recursive inference schema[43], both dragging down forecasting precision. To further boost the performance of the final prediction, many advanced architectures have been introduced to capture the long-range dependencies, such as the self-attention mechanism and the convolutional networks [19, 21, 24]. Besides, to leverage the characteristics of the time series data, recent works also integrate traditional analysis methods like trend-seasonal decomposition and time-frequency conversion into neural networks [36, 44]. In addition, a recent study points out that a simple linear network enhanced with decomposition also achieves competitive performance[39]. Furthermore, slice-based methods demonstrate superior accuracy in the long time series forecasting tasks [26, 41].

### 2.2  Non-stationary Time Series Forecasting

Most time series forecasting methods prioritize designing powerful architectures that can effectively capture temporal dependencies, but often overlook the non-stationary nature of the data. Considering the basic assumption of deep-learning-based models that the data in both training and test sets follow the same distribution, such a discrepancy will definitely drag down the precision of the model for future time prediction. Moreover, the distribution differences among instances in the training set may introduce noise, making the learning task harder to converge. To address these challenges, various stationarization methods have been explored.

In detail, DDG-DA[22] predicts the evolving data distribution in a domain adaptation fashion. Du et al.[8] propose an adaptive RNN to alleviate the impact of non-stationary factors by distribution characterization and distribution matching. Besides, normalization-based approaches have also gained popularity as they aim to remove non-stationary factors from original data and normalize all data to a consistent distribution. DAIN [28] introduces a non-linear network to learn how to normalize each input instance adaptively and ST-norm [7] proposes two normalization modules from both temporal and spatial perspectives. Later researchers point out that non-stationary factors are essential in accurate forecasting and simply removing them may result in poor prediction. Therefore, they propose RevIN [17], a symmetric normalization method that first normalizes the input sequences and then denormalizes the model output sequences through instance normalization [34]. Based on the similar structure, Non-stationary Transformers [25] presents de-stationary attention that incorporates the non-stationary factors in self-attention, resulting in significant improvements over Transformer-based models. Moreover, a recent study [10] identifies the intra- and inter-space distribution shift in time series, and proposes to relieve these issues by learning the distribution coefficients.

Despite the effectiveness of existing normalization methods, they inappropriately assume that all time points within the same instance share the same statistical properties during the normalization processing. Different from them, our proposed approach focuses on further thinking of the nature of the data, i.e., the distribution is inconsistent across compact time slices and such inconsistency is not just on a per-instance basis.

# 3 Proposed Method

We propose a general model-agnostic normalization framework for time series forecasting called Slicing Adaptive Normalization (SAN) to address the inconsistency mentioned above. Considering an input set of time series $X = \{\boldsymbol{x}^i\}_{i=1}^N$ and their horizon series $Y = \{\boldsymbol{y}^i\}_{i=1}^N$, SAN is expected to remove the non-stationary factors and assist the forecasting models to predict more accurately based on the observed input series. In this section, we will present the detailed workflow of the entire framework and explain how it works with non-stationary time series data. To provide better clarity, we summarize the key notations in Table 1 and the whole framework can be referred to in Fig. 2.

Table 1: The Key Mathematical Notations.

| Notation | Description |
|---|---|
| $N$ | the number of instances |
| $V$ | the number of variables |
| $T$ | the given slicing time span |
| $L_{in}, L_{out}$ | the sequence length of input/target sequence |
| $M, K$ | the number of slices of input/target sequence, $M = \frac{L_{in}}{T}, K = \frac{L_{out}}{T}$ |
| $\boldsymbol{x}^i, \boldsymbol{y}^i$ | the i-th input/target series for the whole framework |
| $\bar{\boldsymbol{x}}^i, \bar{\boldsymbol{y}}^i$ | the i-th input/target series for forecasting models |
| $*_j^i$ | the property of j-th slice in the i-th series, determined by * |
| $\mu, \sigma$ | the mean and the standard deviation value |
| $\hat{*}$ | the predicted value, determined by * |

## 3.1 Normalization

Similar to existing normalization methods for non-stationary time series forecasting [17], SAN first normalizes the input sequence to remove the non-stationary factors and later restores them to the output sequence by denormalization. Differently, SAN applies such operation on a per-slice basis instead of the whole input sequence. Such a localized operation can better maintain the instinct pattern of each slice than global instance normalization. The framework first splits the input $\boldsymbol{x}^i$ into $M$ non-overlapping slices $\{\boldsymbol{x}_j^i\}_{j=1}^M$ based on $T$. Then the mean and standard deviation for each slice is computed as:

$$\mu_j^i = \frac{1}{T}\sum_{t=1}^T \boldsymbol{x}_{j,t}^i, (\sigma_j^i)^2 = \frac{1}{T}\sum_{t=1}^T(\boldsymbol{x}_{j,t}^i - \mu_j^i)^2, \tag{1}$$

where $\mu_j^i, \sigma_j^i \in R^{V*1}$ and $\boldsymbol{x}_{j,t}^i$ is the value of slice $\boldsymbol{x}_j^i$ at $t$-th time step. Later, SAN normalizes every slice of the original input sequence by their individual statistics as:

$$\bar{\boldsymbol{x}}_j^i = \frac{1}{\sigma_j^i + \epsilon} \cdot (\boldsymbol{x}_j^i - \mu_j^i). \tag{2}$$

Here we use $\cdot$ to denote the element-wise product and $\epsilon$ is a small constant. Finally, SAN restores all the slices in their original chronological order and lets the processed series without non-stationary factors $\bar{\boldsymbol{x}}^i$ be the new input of the forecasting models.

## 3.2 Statistics Prediction

As illustrated in Fig. 2, SAN introduces a unique statistics prediction module, $f_\phi(*)$, to better estimate future distributions in addition to the backbone forecasting model, $g_\theta(*)$. Unlike existing works that denormalize the entire output of backbone models with the statistics of original input sequences, SAN faces a natural challenge of per-slice normalization: how to **estimate the evolving distributions** for each future slice. To simplify and improve efficiency, we use a two-layer perceptron network with an appropriate activation function (e.g., Relu() for standard deviation to ensure non-negativity) that learns to predict future distributions based on input statistics and stationarized sequence.

The quality of statistics predictions determines the overall performance of SAN since we depend on an accurate estimation of future distribution to restore the non-stationary nature of each instance. In

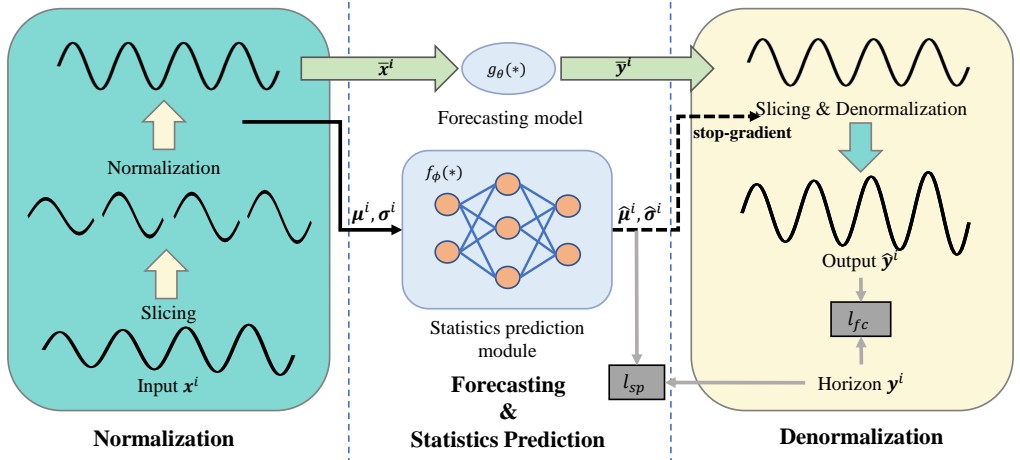

Figure 2: The illustration of the proposed SAN framework. SAN is a model-agnostic symmetrical normalization framework that removes and restores the non-stationary factors in the time series data from the perspective of slicing. SAN trains in a two-stage manner. It first optimizes the statistics prediction module into convergence ($l_{sp}$), which learns to predict future statistics based on sliced input mean $\boldsymbol{\mu}^i$ and standard deviation $\boldsymbol{\sigma}^i$. The second stage is the traditional procedure of training forecasting models ($l_{fc}$), where the statistics prediction module is frozen and functions as a plugin.

our modeling of distribution, the mean determines the approximate scale of a given slice and the standard deviation represents the degree of dispersion, where the scale of a small slice may be more important in the task of forecasting. Therefore, we aim to further refine the modeling approach for the mean component on the basis of analyzing its properties.

In detail, we believe that the overall mean of the input sequence $\rho^i = \frac{1}{L_{in}} \sum_1^{L_{in}} \boldsymbol{x}^i \in R^{V*1}$ is a *maximum likelihood estimation* of the target sequence's mean $\hat{\rho}^i = \frac{1}{L_{out}} \sum_1^{L_{out}} \boldsymbol{y}^i \in R^{V*1}$ since they are temporally related. That is, $\rho^i \approx \hat{\rho}^i$. Such property is widely accepted in existing works [17, 25] as they denormalize the output using the statistics of the whole input sequence. Based on the above assumption, we introduce a ***residual learning*** [13] technique in our method, letting the module learn the difference between the future slice mean $\hat{\boldsymbol{\mu}}^i$ and the overall input mean $\rho^i$, instead of predicting the specific values. This approach reduces the difficulty in modeling means with prior knowledge about future trends. Additionally, to account for different variables exhibiting distinct patterns in scale changes, we further use two learnable vectors $\boldsymbol{W}_1, \boldsymbol{W}_2 \in R^V$ initialized as ones-vector to present the ***individual preference*** weights for each variable, making the prediction computed in a weighted-sum manner. The statistics prediction procedure can be formulated as:

$$\hat{\boldsymbol{\mu}}^i = \boldsymbol{W}_1 * \text{MLP}(\boldsymbol{\mu}^i - \rho^i, \bar{\boldsymbol{x}}^i - \rho^i) + \boldsymbol{W}_2 * \rho^i, \hat{\boldsymbol{\sigma}}^i = \text{MLP}(\boldsymbol{\sigma}^i, \bar{\boldsymbol{x}}^i). \tag{3}$$

Here $\boldsymbol{\mu}^i = [\mu_1^i, \mu_2^i ... \mu_M^i] \in R^{V*M}$ denotes all the mean values of $M$ slices of the input, and $\hat{\boldsymbol{\mu}}^i \in R^{V*K}$ stands for the predicted mean of future $K$ slices. The same notation works for the standard deviation. The mean squared error (MSE) between predicted statistics and ground truth is utilized as the loss function ($l_{sp}$) to train the network through backpropagation.

In this work, we mainly focus on proposing and modeling the non-stationary nature of time series from a slicing perspective. The challenge of how to design powerful deep models for statistics prediction is left for future explorations.

### 3.3 Denormalization

Simultaneous with statistic prediction, SAN feeds the normalized sequence into the forecasting model, which is responsible for producing internal output $\bar{\boldsymbol{y}}^i$. Finally, SAN denormalizes the output given by the backbone, restoring the non-stationary factors for an accurate forecasting result.

Symmetrically, SAN performs on a per-slice basis as illustrated in Fig. 2. For the internal output $\bar{\boldsymbol{y}}^i$, we first split it into $K$ non-overlapping slices $\{\bar{\boldsymbol{y}}_j^i\}_{j=1}^K$. Then the denormalization operation for an arbitrary slice based on our predicted statistics can be defined as the following formula:

$$\hat{\boldsymbol{y}}_j^i = \bar{\boldsymbol{y}}_j^i * (\hat{\boldsymbol{\sigma}}_j^i + \epsilon) + \hat{\boldsymbol{\mu}}_j^i. \tag{4}$$

Finally, by restoring all the slices in their chronological order, we can get the final prediction $\hat{\boldsymbol{y}}^i$ of the whole framework, which will be later used for loss computation ($l_{fc}$) and performance evaluation.

### 3.4 Two-stage Training Schema

Though the overall framework is simple and clear, we find that the training process needs to be carefully deliberated. Since the normalization flow of SAN functions as a constraint to the backbone model, the overall learning procedure is actually a *bi-level optimization problem* [12]. The goal of the upper level is the performance of time series forecasting while the goal of the lower level is the distribution similarity between denormalized output and ground truth. Formally, the original overall training process can be described as:

$$\arg\min_\theta \sum_{(\boldsymbol{x}^i, \boldsymbol{y}^i)} l_{fc}(\theta, \phi^*, (\boldsymbol{x}^i, \boldsymbol{y}^i)),$$
$$\text{s.t.} \phi^* = \arg\min_\phi \sum_{(\boldsymbol{x}^i, \boldsymbol{y}^i)} l_{sp}(\theta, \phi, (\boldsymbol{x}^i, \boldsymbol{y}^i)). \tag{5}$$

Here we omit the transformation process of the data and only keep the original input required for the calculation for brevity.

We propose a two-stage training paradigm for SAN by simplifying the lower-level optimization objective so that it can focus on estimating the future distribution, instead of reducing the distribution discrepancy between the denormalized output of a certain model and the ground truth. Specifically, we optimize $\phi^* = \arg\min_\phi \sum_{(\boldsymbol{x}^i, \boldsymbol{y}^i)} l_{sp}(\phi, (\boldsymbol{x}^i, \boldsymbol{y}^i))$ using stochastic gradient descent. This decouples the original non-stationary forecasting task into a statistic prediction task and a stationary forecasting task. In practice, the statistics prediction module is first trained into convergence, which is then frozen and treated as a plugin during the second stage of training the forecasting model. The training algorithm is provided in the Appendix C.2.

Such a solution has some desirable qualities: The first is simplicity. The two-stage schema allows for a concise and easy-to-follow design of the model architecture and training process. The second is effectiveness. The statistics prediction module is expected to produce reliable predictions on future distribution since it is optimized on the whole training set into convergence. Therefore the forecasting model can handle the simpler task of learning the scale-free pattern in the normalized data. These two advantages greatly meet our ultimate goal of designing a concise yet effective framework for non-stationary time series forecasting tasks. The third and the most important is flexibility. Though there exist many advanced methods for the bi-level optimization problem [5, 11], their upper and lower objectives are always highly related. In contrast, our proposal completely decouples these parts, making SAN a model-agnostic framework that can migrate to various scenarios without special design and further tuning.

## 4 Experiments

In this section, we conduct sufficient experiments within a widely used benchmark dataset compared to state-of-the-art methods to evidence the effectiveness of our proposed SAN framework.

### 4.1 Experimental Setup

**Datasets** We use nine datasets in our experiment and here are brief descriptions of them. (1) **ETT**[3] [43] records the oil temperature and load features of the electricity transformers from July 2016 to July 2018. It is made up of 4 sub datasets where ETThs are sampled per hour and ETTms are

---

[3]https://github.com/zhouhaoyi/ETDataset

sampled every 15 minutes. (2) **Electricity**[4] contains the electricity consumption data of 321 clients from July 2016 to July 2019. (3) **Exchange**[5] [19] collects the daily exchange rates of 8 countries from 1990 to 2016. (4) **Traffic**[6] includes the hourly traffic load of San Francisco freeways recorded by 862 sensors from 2015 to 2016. (5) **Weather**[7] is made up of 21 indicators of weather, including air temperature and humidity collected every 10 minutes in 2021. (6) **ILI**[8] records the weekly ratio of influenza-like illness patients versus the total patients by the Centers for Disease Control and Prevention of the United States from 2002 to 2021. The detailed information about these datasets are listed in the Table. 2. We also report the ADF test (Augmented Dickey-Fuller Test) [9] results in the table, which evaluate the stationarity of a time series. Following the standard protocol, we split each dataset into training, validation and testing sets according to the chronological order. The split ratio is 6:2:2 for ETT dataset and 7:1:2 for the other datasets [38]. Also, we apply a z-score normalization on them based on the statistics of training data as preprocessing to measure different variables on the same scale. Note that z-score normalization is unable to handle non-stationary time series since the statistics are fixed during normalization [28].

Table 2: The Statistics of Each Dataset.

| Dataset | Variables | Sampling Frequency | Length | Slicing Length | ADF* |
|---|---|---|---|---|---|
| Electricity | 321 | 1 Hour | 26,304 | 24 | -8.44 |
| Exchange | 8 | 1 Day | 7,588 | 6 | -1.90 |
| Traffic | 862 | 1 Hour | 17,544 | 24 | -15.02 |
| Weather | 21 | 10 Minutes | 52,696 | 12 | -26.68 |
| ILI | 7 | 1 Week | 966 | 6 | -5.33 |
| ETTh1&ETTh2 | 7 | 1 Hour | 17,420 | 24 | -5.91&-4.13 |
| ETTm1&ETTm2 | 7 | 15 Minutes | 69,680 | 12 | -14.98&-5.66 |

*A smaller ADF test result indicates a more stationary time series data

**Backbone models** SAN is a model-agnostic framework that can be applied to arbitrary time series forecasting models. To evidence the effectiveness of the framework, we select some mainstream models based on different architectures and evaluate their performance under both multivariate and univariate settings: Linear model based **DLinear** [39], Transformer based **Autoformer** [38] and **FEDformer** [45], and dilated convolution based **SCINet** [24]. We follow the implementation and settings provided in the official code of DLinear[9] and SCINet[10] to implement these models.

**Experiments details** We use ADAM [18] as the default optimizer across all the experiments and report the mean squared error (MSE) and mean absolute error (MAE) as the evaluation metrics. A lower MSE/MAE indicates a better performance. For the statistics prediction module in SAN, we use a simple two-layer perceptron network with a hidden size the same as the embedding size of the backbone model for simplicity. The detailed implementation of the statistics prediction module can be referred to in Appendix C.1. All the experiments are implemented by PyTorch [29] and are conducted for three runs with a fixed random seed on a single NVIDIA RTX 3090 24GB GPU.

**Slicing length** Regarding the selection of slicing length for each dataset, we adopt a heuristic idea that real-world time series data exhibit similar changing patterns within artificially defined or actual periods (daily, weekly, etc.). Combing the frequencies of benchmark datasets, we establish a range of $\{6, 12, 24, 48\}$ as slicing lengths such that most settings cover a meaningful time span. For example, we selected a slicing length of 24 for datasets such as ETTh1, Electricity and Traffic with a frequency of 1 hour. This ensures that each time slice contained data within a day and guarantees optimal performance among candidates. Here we admit that one limitation of our method is that the current design cannot handle indivisible length such that we set the slicing length to 6 which approximately represents a weekly period instead of 7 in the Exchange dataset. We present the ablation study on the effect of slicing length in Appendix B.5.

---

[4]https://archive.ics.uci.edu/ml/datasets/ElectricityLoadDiagrams20112014

[5]https://github.com/laiguokun/multivariate-time-series-data

[6]http://pems.dot.ca.gov

[7]https://www.bgc-jena.mpg.de/wetter/

[8]https://gis.cdc.gov/grasp/fluview/fluportaldashboard.html

[9]https://github.com/cure-lab/LTSF-Linear

[10]https://github.com/cure-lab/SCINet

Table 3: Forecasting errors under the multivariate setting. The **bold** values indicate better performance.

| Methods Metric | Dlinear MSE | MAE | + SAN MSE | MAE | FEDformer MSE | MAE | + SAN MSE | MAE | Autoformer MSE | MAE | + SAN MSE | MAE | SCINet MSE | MAE | + SAN MSE | MAE |
|---|---|---|---|---|---|---|---|---|---|---|---|---|---|---|---|---|
| **Electricity** | | | | | | | | | | | | | | | | |
| 96 | 0.140 | 0.237 | **0.137** | **0.234** | 0.185 | 0.300 | **0.164** | **0.272** | 0.195 | 0.309 | **0.172** | **0.281** | 0.213 | 0.316 | **0.152** | **0.256** |
| 192 | 0.153 | 0.250 | **0.151** | **0.247** | 0.196 | 0.310 | **0.179** | **0.286** | 0.215 | 0.325 | **0.195** | **0.300** | 0.224 | 0.329 | **0.163** | **0.266** |
| 336 | 0.168 | 0.267 | **0.166** | **0.264** | 0.215 | 0.330 | **0.191** | **0.299** | 0.237 | 0.344 | **0.211** | **0.316** | 0.230 | 0.334 | **0.178** | **0.283** |
| 720 | 0.203 | 0.301 | **0.201** | **0.295** | 0.244 | 0.352 | **0.230** | **0.334** | 0.292 | 0.375 | **0.236** | **0.335** | 0.260 | 0.356 | **0.206** | **0.307** |
| **Exchange** | | | | | | | | | | | | | | | | |
| 96 | 0.086 | **0.213** | **0.085** | 0.214 | 0.152 | 0.281 | **0.079** | **0.205** | 0.152 | 0.283 | **0.082** | **0.208** | 0.126 | 0.269 | **0.082** | **0.200** |
| 192 | **0.161** | **0.297** | 0.177 | 0.317 | 0.273 | 0.380 | **0.156** | **0.295** | 0.369 | 0.437 | **0.157** | **0.296** | 0.266 | 0.392 | **0.169** | **0.293** |
| 336 | 0.338 | 0.437 | **0.294** | **0.407** | 0.452 | 0.498 | **0.260** | **0.384** | 0.534 | 0.544 | **0.262** | **0.385** | 0.574 | 0.541 | **0.320** | **0.409** |
| 720 | 0.999 | 0.755 | **0.726** | **0.649** | 1.151 | 0.830 | **0.697** | **0.633** | 1.222 | 0.848 | **0.689** | **0.629** | 1.136 | 0.818 | **0.892** | **0.712** |
| **Traffic** | | | | | | | | | | | | | | | | |
| 96 | **0.411** | **0.283** | 0.412 | 0.288 | 0.579 | 0.363 | **0.536** | **0.330** | 0.654 | 0.403 | **0.569** | **0.350** | 0.626 | 0.393 | **0.542** | **0.344** |
| 192 | **0.423** | **0.289** | 0.429 | 0.297 | 0.608 | 0.376 | **0.565** | **0.345** | 0.654 | 0.410 | **0.594** | **0.364** | 0.613 | 0.396 | **0.545** | **0.358** |
| 336 | **0.437** | **0.297** | 0.445 | 0.306 | 0.620 | 0.385 | **0.580** | **0.354** | 0.629 | 0.391 | **0.591** | **0.363** | 0.625 | 0.398 | **0.563** | **0.369** |
| 720 | **0.467** | **0.316** | 0.474 | 0.319 | 0.630 | 0.387 | **0.607** | **0.367** | 0.657 | 0.402 | **0.623** | **0.380** | 0.639 | 0.409 | **0.607** | **0.381** |
| **Weather** | | | | | | | | | | | | | | | | |
| 96 | 0.175 | 0.237 | **0.152** | **0.210** | 0.246 | 0.328 | **0.179** | **0.239** | 0.247 | 0.320 | **0.194** | **0.256** | 0.181 | 0.260 | **0.169** | **0.232** |
| 192 | 0.217 | 0.275 | **0.196** | **0.254** | 0.281 | 0.341 | **0.234** | **0.296** | 0.302 | 0.361 | **0.258** | **0.316** | 0.239 | 0.311 | **0.215** | **0.275** |
| 336 | 0.263 | 0.314 | **0.246** | **0.294** | 0.337 | 0.376 | **0.304** | **0.348** | 0.362 | 0.394 | **0.329** | **0.367** | 0.293 | 0.348 | **0.267** | **0.314** |
| 720 | 0.325 | 0.366 | **0.315** | **0.346** | 0.414 | 0.426 | **0.400** | **0.404** | **0.427** | **0.433** | 0.440 | 0.438 | 0.345 | 0.380 | **0.338** | **0.365** |
| **ILI** | | | | | | | | | | | | | | | | |
| 24 | 2.297 | 1.055 | **2.122** | **1.001** | 3.205 | 1.255 | **2.614** | **1.119** | 3.309 | 1.270 | **2.777** | **1.157** | 7.467 | 2.039 | **2.776** | **1.163** |
| 36 | 2.323 | 1.070 | **2.029** | **0.978** | 3.148 | 1.288 | **2.537** | **1.079** | 3.207 | 1.216 | **2.649** | **1.104** | 7.035 | 1.948 | **2.411** | **1.026** |
| 48 | 2.262 | 1.065 | **2.041** | **0.971** | 2.913 | 1.168 | **2.416** | **1.032** | 3.166 | 1.198 | **2.420** | **1.029** | 7.225 | 1.955 | **2.295** | **1.004** |
| 60 | 2.443 | 1.124 | **2.089** | **0.973** | 2.853 | 1.161 | **2.299** | **1.003** | 2.947 | 1.159 | **2.401** | **1.021** | 7.335 | 1.957 | **2.487** | **1.063** |
| **ETTh2** | | | | | | | | | | | | | | | | |
| 96 | 0.292 | 0.356 | **0.277** | **0.338** | 0.341 | 0.382 | **0.300** | **0.355** | 0.384 | 0.420 | **0.316** | **0.366** | 0.690 | 0.625 | **0.294** | **0.347** |
| 192 | 0.383 | 0.418 | **0.340** | **0.378** | 0.426 | 0.436 | **0.392** | **0.413** | 0.457 | 0.454 | **0.413** | **0.426** | 0.991 | 0.742 | **0.374** | **0.398** |
| 336 | 0.473 | 0.477 | **0.356** | **0.398** | 0.481 | 0.479 | **0.459** | **0.462** | 0.468 | 0.473 | **0.446** | **0.457** | 1.028 | 0.759 | **0.412** | **0.430** |
| 720 | 0.708 | 0.599 | **0.396** | **0.435** | **0.458** | 0.477 | 0.462 | **0.472** | 0.473 | 0.485 | **0.471** | **0.474** | 1.363 | 0.885 | **0.437** | **0.461** |

Table 4: Comparison between SAN and existing normalization approaches. The best results are highlighted in **bold**.

| Methods | FEDformer +SAN | +RevIN | +NST | +Dish-TS | IMP(%) | Autoformer +SAN | +RevIN | +NST | +Dish-TS | IMP(%) |
|---|---|---|---|---|---|---|---|---|---|---|
| Electricity | **0.191** | 0.200 | 0.198 | 0.203 | 3.54 | **0.204** | 0.219 | 0.213 | 0.231 | 4.23 |
| Exchange | **0.298** | 0.474 | 0.480 | 0.704 | 37.13 | **0.297** | 0.495 | 0.494 | 1.008 | 39.88 |
| Traffic | **0.572** | 0.647 | 0.649 | 0.652 | 11.59 | **0.594** | 0.666 | 0.664 | 0.677 | 10.54 |
| Weather | 0.279 | 0.268 | **0.267** | 0.398 | -4.49 | 0.305 | **0.290** | **0.290** | 0.433 | -5.17 |
| ILI | **2.467** | 2.962 | 3.084 | 2.846 | 13.32 | **2.562** | 3.151 | 3.235 | 3.180 | 18.69 |
| ETTh1 | **0.447** | 0.463 | 0.456 | 0.461 | 1.97 | **0.518** | 0.519 | 0.521 | 0.521 | 0.19 |
| ETTh2 | **0.404** | 0.465 | 0.481 | 1.004 | 13.12 | **0.411** | 0.489 | 0.465 | 1.175 | 11.61 |
| ETTm1 | **0.377** | 0.415 | 0.411 | 0.422 | 8.27 | **0.406** | 0.562 | 0.535 | 0.567 | 24.11 |
| ETTm2 | **0.287** | 0.310 | 0.315 | 0.759 | 7.42 | **0.311** | 0.325 | 0.331 | 0.894 | 4.31 |

## 4.2 Main Results

We report the multivariate forecasting results in Table 3. The ILI dataset has a forecasting horizon of $L_{out} \in \{24, 36, 48, 60\}$ while the others have a forecasting horizon of $L_{out} \in \{96, 192, 336, 720\}$. As for the input sequence length, we follow the traditional protocol and fix $L_{in} = 96$ for Autoformer, FEDformer and SCINet with respect to all datasets ($L_{in} = 36$ for ILI dataset) and extend it to 336 (96 for ILI dataset) for DLinear. Full benchmarks of ETT datasets and univariate results are provided in the Appendix.

As shown in the table, we clearly find that our proposed SAN framework can boost these models by a large margin in most cases of the benchmark dataset. We attribute this improvement to two aspects. Firstly, SAN mitigates the impact of non-stationary factors, as demonstrated by the performance on three typical non-stationary datasets (Exchange, ILI and ETTh2, determined by ADF test results). Specifically, under all experimental forecasting lengths with DLinear, SAN achieves an average MSE reduction of **7.67%** in the Exchange dataset, **11.13%** in the ILI dataset and **21.29%** in the ETTh2 dataset. This conclusion applies to other backbone models as well and the enhancement is even more pronounced. Secondly, even in long-term forecasting scenarios where the difficulty of forecasting increases significantly with the length of the forecast, SAN imposes constraints on

backbone models to produce more reliable results using a novel statistical prediction module. For instance, when predicting for a length of 720 time steps, SCINet accompanied by SAN achieves a **70.37%** reduction in MSE on the ETTh2 dataset and a **20.77%** reduction on the Electricity dataset. These improvements make SCINet comparable to other forecasting models and suggest that SAN can help stabilize outputs in long-term forecasting scenarios.

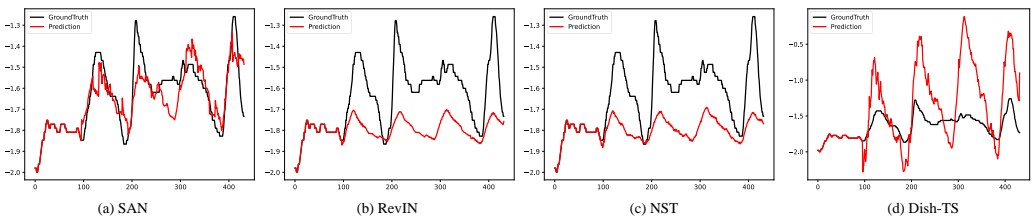

Figure 3: Visualization of long-term forecasting results of a sample of ETTm2 dataset given by FEDformer enhanced with different normalization methods.

### 4.3   Comparison With Normalization Methods

In this section, we compare SAN with three state-of-the-art normalization methods for non-stationary time series forecasting: **RevIN** [17], Non-Stationary Transformers (**NST**) [25] and **Dish-TS** [10]. Following the same experimental settings in Section 4.2, we report the average MSE evaluation of Autoformer and FEDformer over all the forecasting lengths for each dataset and the relative improvements in Table 4. Other models are not as involved backbone since NST can only suit Transformer-based models, while the rest methods are much more flexible to be applied for arbitrary forecasting models on the contrary.

It can be concluded that SAN achieves the best performance among existing normalization methods. The improvement is significant with an average MSE decrease of **10.71%** by FEDformer. SAN consistently performs better than baseline models except for Weather and the improvement is more pronounced in typical non-stationary datasets like Exchange and ILI (determined by the ADF test). The comparison reveals that SAN may be more effective at removing non-stationary factors from a time-slicing perspective rather than considering the entire instance. Additionally, the proposed two-stage training schema is crucial as it enables SAN to outperform Dish-TS by a large margin, which ignores the bi-level optimization nature. However, this exceptional ability of SAN may lead to an over-stationarization problem [25], resulting in decreased performance on the Weather dataset. The detailed results of all cases and further discussions are provided in Appendix B.6.

### 4.4   Qualitative Evaluation

The quality of prediction results in time series forecasting is crucial, in addition to the accuracy of metrics. Figure 3 displays a sample forecast on the ETTm2 dataset using FEDformer as the backbone with SAN, RevIN, NST or Dish-TS enhancements. The input length is 96 and the forecasting length is set to 336. It's evident that SAN produces more realistic predictions while its counterparts even fail to capture the scale of future data. We guess the poor quality of RevIN and NST is caused by their coarse way of denormalizing. Although the mean value of an input sequence can be considered a maximum likelihood estimation for future data, it's likely that non-stationary datasets' distribution will change significantly in comparison to inputs. Therefore, simply denormalizing output from backbone models with input sequence statistics may lead to mismatches like those seen in RevIN and NST forecasts where both scales are similar. As for Dish-TS, though the method tries to learn future distribution, it ignores the bi-level optimization nature and its entangled learning schema limits the estimation accuracy of statistics and finally leads to poor performance. On the opposite, SAN models the dynamic nature of time series from a slicing perspective and introduces an independent statistics prediction module to learn to predict the future distribution for denormalizing by a two-stage training schema. In this way, we adaptively adjust the scale and bias of forecasting results based on the statistic predictions, capturing the tendency of future data. As a result, though the average value of the input is rather low, SAN still produces higher predictions that are consistent with ground truth.

# 5 Conclusion

In this study, we focused on alleviating the non-stationary property of time series data using a novel slice view. We proposed the SAN framework for time series forecasting, which is a model-agnostic approach that normalizes the input by removing non-stationary factors and restores them to the output through denormalization on a per-slice basis. Additionally, with the help of a novel statistics prediction module, SAN simplifies non-stationary forecasting by dividing it into two subtasks to improve forecasting model performance. To demonstrate the superiority of SAN, we conducted experiments on a widely used benchmark dataset and found that SAN significantly improves mainstream forecasting models and outperforms state-of-the-art normalization methods. We hope that SAN can serve as a foundation component for time series forecasting, and stimulate further research on modeling time series from a slice perspective.

# 6 Acknowledgement

This research was partially supported by grants from the National Natural Science Foundation of China (Grant No. U20A20229). This work also thanked to the support of funding MAI2022C007. We furthermore thanked the anonymous reviewers for their constructive comments.

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

# A  Effects of SAN on Non-stationary Time Series Forecasting

## A.1  Discussions

As illustrated in the main paper, our proposed SAN is a compact plug-and-play framework. We will first give a brief discussion in this section on how SAN can be effective.

It is of utmost importance that SAN can well alleviate the impact of the non-stationary nature of time series data. Forecasting models may encounter a non-i.i.d problem with non-stationary data, that is, the marginal distribution of each input instance can be different, which may lead to a huge difference between the distribution of the training set and the test set. Thus the models can not generalize well in future predictions. However, SAN will normalize all the input instances into a standard normal distribution and force the mean and variance of the training and test data distributions to be identical. In this way, all the data instances are from the same distribution, therefore the forecasting task is simplified as the models can get rid of the noises caused by non-stationary factors and only focus on mining the time-invariant patterns. Moreover, compared to existing normalization methods for forecasting, our modeling of the non-stationary property in a time slice view is more in-depth and realistic, so SAN can better remove the non-stationary factors in input sequences while keeping their instinct information in the normalization phase. Hence, SAN is theoretically expected to perform better in non-stationary time series forecasting.

Another part that contributes to the effectiveness of SAN is the statistics prediction module and the two-stage training schema. With the statistics prediction module independently modeling the evolving trends of statistical properties, SAN adopts more precise statistics for adaptive denormalization than existing solutions. Moreover, the proposed two-stage strategy actually simplifies the original forecasting task by divide and conquer: In the first stage we try to learn the general direction and dispersion of the future data, which is easy to fit and is conducted by the light statistics prediction module. Next, we utilize the powerful backbone model to discover the scale-free periodic-like features to estimate future values under the guidance of the well-trained statistics prediction module. Therefore, backbone models in SAN are actually responsible for an easier subtask. Considering that SAN can usually give reliable estimations on future distributions, SAN is expected to perform well on non-stationary time series forecasting by splitting the task into two simpler subtasks.

## A.2  Theoretical Analysis

Using the same notation in the paper, we prove that all the inputs after SAN's normalization follow a standard normal distribution, validating SAN's capability to remove the non-stationary factors theoretically.

In detail, for arbitrary input sequence $\boldsymbol{x}^i$, SAN first split it into $M$ non-overlapping slices $\{\boldsymbol{x}_j^i\}_{j=1}^M$ and normalizes them according to their statistics. Therefore we will get:

$$\forall i, j \; \mathbb{E}[\bar{\boldsymbol{x}}_j^i] = 0, Var[\bar{\boldsymbol{x}}_j^i] = I \tag{6}$$

And as for the statistics of normalized input $\bar{\boldsymbol{x}}^i$, it satisfies the following equations:

$$
\begin{aligned}
\mathbb{E}[\bar{\boldsymbol{x}}^i] &= \mathbb{E}_j[\mathbb{E}[\bar{\boldsymbol{x}}_j^i]] \\
&= \mathbb{E}_j[0] \\
&= 0
\end{aligned}
\tag{7}
$$

$$
\begin{aligned}
Var[\bar{\boldsymbol{x}}^i] &= \frac{\sum_{t=0}^{L_{in}} (\bar{x}_{:,t}^i - \mathbb{E}[\bar{\boldsymbol{x}}^i])^2}{L_{in}} \\
&= \frac{\sum_{t=0}^{L_{in}} (\bar{x}_{:,t}^i)^2}{MT} \\
&= \frac{1}{M} * \left( \frac{\sum_{t=0}^{T} (\bar{x}_{:,t}^i)^2}{T} + \frac{\sum_{t=T}^{2T} (\bar{x}_{:,t}^i)^2}{T} + , \ldots, \frac{\sum_{t=(M-1)T}^{MT} (\bar{x}_{:,t}^i)^2}{T} \right) \\
&= \mathbb{E}_j[Var[\bar{\boldsymbol{x}}_j^i]] \\
&= I
\end{aligned}
\tag{8}
$$

Here $\bar{x}^i_{:,t} \in R^{V*1}$ denotes all the normalized variables in time step $t$. From the above equations, we can learn that any input sequence follows a standard normal distribution after the normalization operation of SAN, which meets our expectations.

# B Supplementary Experiments

## B.1 Full Benchmark on the ETT Dataset

Table 5: Multivariate forecasting results on full ETT dataset.

| Methods | DLinear | | + SAN | | FEDformer | | + SAN | | Autoformer | | + SAN | | SCINet | | + SAN | |
| Metric | MSE | MAE | MSE | MAE | MSE | MAE | MSE | MAE | MSE | MAE | MSE | MAE | MSE | MAE | MSE | MAE |
|---|---|---|---|---|---|---|---|---|---|---|---|---|---|---|---|---|
| ETTh1 96 | **0.377** | **0.399** | 0.383 | **0.399** | **0.371** | 0.411 | 0.383 | **0.409** | **0.458** | **0.448** | 0.488 | 0.464 | 0.470 | 0.479 | **0.391** | **0.405** |
| ETTh1 192 | **0.417** | 0.426 | 0.419 | **0.419** | **0.420** | 0.443 | 0.431 | **0.438** | **0.481** | 0.474 | 0.498 | **0.472** | 0.541 | 0.520 | **0.438** | **0.433** |
| ETTh1 336 | 0.464 | 0.461 | **0.437** | **0.432** | **0.446** | 0.459 | 0.471 | **0.456** | **0.508** | **0.485** | 0.530 | 0.498 | 0.643 | 0.587 | **0.477** | **0.451** |
| ETTh1 720 | 0.493 | 0.505 | **0.446** | **0.459** | **0.482** | 0.495 | 0.504 | **0.488** | **0.525** | 0.516 | 0.555 | **0.514** | 0.774 | 0.669 | **0.489** | **0.474** |
| ETTh2 96 | 0.292 | 0.356 | **0.277** | **0.338** | 0.341 | 0.382 | **0.300** | **0.355** | 0.384 | 0.420 | **0.316** | **0.366** | 0.690 | 0.625 | **0.294** | **0.347** |
| ETTh2 192 | 0.383 | 0.418 | **0.340** | **0.378** | 0.426 | 0.436 | **0.392** | **0.413** | 0.457 | 0.454 | **0.413** | **0.426** | 0.991 | 0.742 | **0.374** | **0.398** |
| ETTh2 336 | 0.473 | 0.477 | **0.356** | **0.398** | 0.481 | 0.479 | **0.459** | **0.462** | 0.468 | 0.473 | **0.446** | **0.457** | 1.028 | 0.759 | **0.412** | **0.430** |
| ETTh2 720 | 0.708 | 0.599 | **0.396** | **0.435** | **0.458** | 0.477 | 0.462 | **0.472** | 0.473 | 0.485 | **0.471** | **0.474** | 1.363 | 0.885 | **0.437** | **0.461** |
| ETTm1 96 | 0.301 | 0.344 | **0.288** | **0.342** | 0.362 | 0.408 | **0.311** | **0.355** | 0.493 | 0.470 | **0.343** | **0.378** | 0.444 | 0.464 | **0.321** | **0.360** |
| ETTm1 192 | 0.335 | 0.366 | **0.323** | **0.363** | 0.395 | 0.427 | **0.351** | **0.383** | 0.546 | 0.498 | **0.390** | **0.400** | 0.491 | 0.500 | **0.347** | **0.380** |
| ETTm1 336 | 0.370 | 0.387 | **0.357** | **0.384** | 0.441 | 0.454 | **0.390** | **0.407** | 0.658 | 0.543 | **0.415** | **0.418** | 0.572 | 0.556 | **0.385** | **0.403** |
| ETTm1 720 | 0.425 | 0.421 | **0.409** | **0.415** | 0.488 | 0.481 | **0.456** | **0.444** | 0.626 | 0.532 | **0.476** | **0.453** | 0.728 | 0.654 | **0.450** | **0.441** |
| ETTm2 96 | 0.169 | 0.263 | **0.166** | **0.258** | 0.191 | 0.283 | **0.175** | **0.266** | 0.261 | 0.329 | **0.236** | **0.317** | 0.303 | 0.404 | **0.176** | **0.267** |
| ETTm2 192 | 0.232 | 0.310 | **0.223** | **0.302** | 0.261 | 0.326 | **0.246** | **0.315** | 0.282 | 0.339 | **0.260** | **0.329** | 0.568 | 0.569 | **0.240** | **0.311** |
| ETTm2 336 | 0.303 | 0.361 | **0.272** | **0.330** | 0.327 | 0.365 | **0.315** | **0.362** | 0.350 | 0.378 | **0.330** | **0.376** | 0.793 | 0.689 | **0.300** | **0.351** |
| ETTm2 720 | 0.403 | 0.424 | **0.360** | **0.384** | 0.428 | 0.423 | **0.412** | **0.422** | 0.438 | **0.428** | 0.417 | **0.428** | 1.200 | 0.851 | **0.391** | **0.405** |

We provide the full multivariate forecasting results on the ETT dataset in Table 5, which includes the hourly datasets ETTh1&ETTh2 and the 15-minutes datasets ETTm1&ETTm2. It is obvious that SAN also achieves significant improvements on these datasets on various backbone models.

## B.2 Univariate Forecasting Results

Following the same settings of our main experiment, we provide the univariate forecasting results in Table 6. Similar to the results of multivariate forecasting, SAN can boost the performance of mainstream forecasting models in most cases. On average of all the benchmark settings, DLinear enhanced by SAN reduces MSE by **6.04%** (from 0.230 to 0.214). The improvements for FEDformer, Autoformer and SCINet are **15.40%**, **29.27%** and **36.29%** respectively.

## B.3 Validation on Various Input Lengths

The input length plays an essential role in time series forecasting tasks as it determines how much historical temporal information the model can mine. One may hope that for powerful deep models, the longer the input length, the better the forecasting results. However, a recent study on this question reveals that deep Transformer-based models are not capable of capturing temporal dependencies in the long-term input sequences [39]. That is, the performance of these deep models stays stable or even degrades when the input length increases.

Apart from the design of these deep models, we hold that such a phenomenon can be raised by the non-stationary property of time series. As the input length increases, the variance among input sequences grows larger and ultimately makes it harder for deep models to discover the time-invariant patterns. Therefore, by removing the non-stationary factors in the input by SAN, deep models are expected to exhibit a steady decline in metrics with longer input lengths.

To evidence our thoughts, we conduct long-term forecasting experiments, i.e., $L_{out} = 720$, with various input lengths $L_{in} \in \{24, 48, 72, 96, 120, 144, 168, 192, 336, 504, 672, 720\}$ on the Transformer-based models. Here we choose Transformer [35], Informer [43], Autoformer [38] and FEDformer [45] as the backbone models. The MSE evaluations are plotted in Fig. 4. Note that we omit large values in the line chart to better demonstrate the trend of the overall results. From the figure, we can see that with the assistance of SAN, the performance of deep models with long sequence input is

Table 6: Univariate forecasting results. The **bold** values indicate better performance.

| Methods | | DLinear | | + SAN | | FEDformer | | + SAN | | Autoformer | | + SAN | | SCINet | | + SAN | |
|---|---|---|---|---|---|---|---|---|---|---|---|---|---|---|---|---|---|
| Metric | | MSE | MAE | MSE | MAE | MSE | MAE | MSE | MAE | MSE | MAE | MSE | MAE | MSE | MAE | MSE | MAE |
| Electricity | 96 | **0.203** | **0.315** | 0.204 | 0.317 | 0.302 | 0.413 | **0.248** | **0.363** | 0.442 | 0.490 | **0.283** | **0.386** | 0.364 | 0.435 | **0.321** | **0.412** |
| | 192 | **0.233** | **0.336** | 0.238 | 0.341 | 0.377 | 0.459 | **0.278** | **0.379** | 0.555 | 0.550 | **0.296** | **0.393** | 0.345 | 0.419 | **0.328** | **0.412** |
| | 336 | **0.268** | **0.363** | 0.278 | 0.371 | 0.673 | 0.636 | **0.324** | **0.411** | 0.617 | 0.620 | **0.359** | **0.440** | 0.368 | **0.435** | **0.363** | 0.436 |
| | 720 | 0.330 | 0.425 | **0.325** | **0.420** | 0.575 | 0.575 | **0.502** | **0.514** | 0.645 | 0.624 | **0.443** | **0.503** | 0.420 | 0.478 | **0.410** | **0.477** |
| Exchange | 96 | **0.108** | **0.254** | 0.138 | 0.288 | 0.134 | 0.272 | **0.113** | **0.252** | 0.155 | 0.305 | **0.097** | **0.233** | 0.167 | 0.332 | **0.090** | **0.226** |
| | 192 | **0.193** | **0.350** | 0.287 | 0.436 | **0.290** | 0.418 | 0.307 | 0.404 | 0.405 | 0.495 | **0.208** | **0.358** | 0.486 | 0.552 | **0.185** | **0.335** |
| | 336 | 0.428 | **0.511** | **0.416** | 0.523 | 0.490 | 0.542 | **0.431** | **0.501** | 0.874 | 0.728 | **0.401** | **0.495** | 0.579 | 0.608 | **0.396** | **0.484** |
| | 720 | 1.137 | 0.848 | **0.859** | **0.719** | 1.302 | 0.883 | **1.188** | **0.835** | 1.193 | 0.845 | **1.071** | **0.787** | **0.853** | **0.740** | 1.106 | 0.797 |
| Traffic | 96 | 0.124 | **0.197** | **0.123** | 0.199 | 0.179 | 0.282 | **0.144** | **0.236** | 0.265 | 0.375 | **0.172** | **0.273** | 0.352 | 0.430 | **0.267** | **0.364** |
| | 192 | 0.125 | **0.200** | **0.124** | **0.200** | 0.211 | 0.316 | **0.141** | **0.232** | 0.266 | 0.372 | **0.211** | **0.316** | 0.291 | 0.377 | **0.240** | **0.338** |
| | 336 | **0.126** | **0.206** | 0.228 | 0.269 | 0.369 | 0.458 | **0.207** | **0.318** | 0.284 | 0.371 | **0.164** | **0.259** | 0.298 | 0.387 | 0.347 | 0.396 |
| | 720 | 0.141 | 0.226 | **0.138** | **0.223** | **0.300** | **0.407** | 0.477 | 0.526 | 0.260 | 0.369 | **0.179** | **0.286** | 0.339 | 0.417 | **0.311** | **0.384** |
| Weather | 96 | 0.004 | 0.047 | **0.002** | **0.032** | **0.002** | **0.037** | 0.003 | 0.042 | 0.004 | 0.047 | **0.002** | **0.038** | 0.005 | 0.060 | **0.003** | **0.039** |
| | 192 | 0.005 | 0.057 | **0.002** | **0.037** | 0.005 | 0.058 | **0.004** | **0.049** | **0.003** | 0.045 | **0.003** | 0.047 | 0.006 | 0.065 | **0.002** | **0.036** |
| | 336 | 0.006 | 0.068 | **0.003** | **0.047** | **0.003** | **0.045** | 0.004 | 0.052 | 0.008 | 0.068 | **0.003** | **0.046** | 0.007 | 0.068 | **0.004** | **0.049** |
| | 720 | 0.007 | 0.070 | **0.004** | **0.050** | 0.011 | 0.080 | **0.004** | **0.048** | 0.058 | 0.176 | **0.004** | **0.049** | 0.007 | 0.070 | **0.003** | **0.045** |
| ILI | 24 | 0.741 | 0.681 | **0.663** | **0.626** | 0.910 | 0.825 | **0.798** | **0.688** | 0.865 | 0.800 | **0.765** | **0.721** | 6.336 | 2.130 | **0.707** | **0.665** |
| | 36 | 0.570 | 0.634 | **0.552** | **0.599** | 0.873 | 0.823 | **0.697** | **0.691** | 0.984 | 0.855 | **0.660** | **0.693** | 6.159 | 1.998 | **0.743** | **0.706** |
| | 48 | 0.740 | 0.742 | **0.647** | **0.669** | 1.027 | 0.904 | **0.820** | **0.761** | 1.105 | 0.925 | **0.753** | **0.752** | 6.597 | 2.082 | **0.783** | **0.744** |
| | 60 | 0.911 | 0.848 | **0.765** | **0.743** | 1.221 | 1.002 | **0.981** | **0.839** | 1.222 | 0.982 | **1.024** | **0.904** | 7.556 | 2.418 | **0.902** | **0.801** |
| ETTh1 | 96 | 0.058 | **0.180** | **0.056** | 0.181 | 0.097 | 0.241 | **0.067** | **0.195** | 0.093 | 0.241 | **0.062** | **0.188** | 0.110 | 0.262 | **0.057** | **0.180** |
| | 192 | 0.078 | 0.216 | **0.076** | **0.212** | 0.109 | 0.257 | **0.081** | **0.215** | 0.121 | 0.290 | **0.082** | **0.216** | 0.152 | 0.312 | **0.075** | **0.209** |
| | 336 | 0.099 | 0.246 | **0.092** | **0.240** | 0.103 | 0.251 | **0.098** | **0.240** | 0.115 | 0.271 | **0.089** | **0.232** | 0.183 | 0.350 | **0.093** | **0.238** |
| | 720 | 0.158 | 0.322 | **0.092** | **0.240** | 0.130 | 0.290 | **0.103** | **0.248** | 0.108 | 0.259 | **0.106** | **0.249** | 0.252 | 0.432 | **0.096** | **0.245** |
| ETTh2 | 96 | **0.132** | **0.280** | 0.133 | 0.281 | 0.145 | 0.301 | **0.141** | **0.286** | 0.181 | 0.332 | **0.141** | **0.288** | 0.149 | 0.306 | **0.129** | **0.274** |
| | 192 | 0.177 | 0.330 | **0.174** | **0.327** | 0.188 | 0.339 | **0.184** | **0.331** | 0.213 | 0.371 | **0.196** | **0.350** | 0.187 | 0.340 | **0.178** | **0.326** |
| | 336 | 0.207 | 0.366 | **0.200** | **0.359** | 0.220 | 0.380 | 0.224 | 0.371 | 0.232 | 0.391 | **0.221** | **0.370** | 0.236 | 0.385 | **0.222** | **0.374** |
| | 720 | 0.301 | 0.447 | **0.237** | **0.391** | 0.279 | 0.427 | **0.257** | **0.407** | 0.267 | 0.417 | **0.289** | **0.431** | 0.326 | 0.468 | **0.272** | **0.421** |
| ETTm1 | 96 | 0.027 | **0.123** | **0.026** | **0.123** | 0.060 | 0.193 | **0.028** | **0.125** | 0.059 | 0.193 | **0.027** | **0.125** | 0.065 | 0.204 | **0.032** | **0.135** |
| | 192 | 0.045 | 0.156 | **0.040** | **0.151** | 0.065 | 0.202 | **0.044** | **0.159** | 0.083 | 0.231 | **0.042** | **0.155** | 0.198 | 0.342 | **0.049** | **0.168** |
| | 336 | 0.059 | 0.178 | **0.055** | **0.176** | 0.066 | 0.199 | **0.059** | **0.189** | 0.069 | 0.205 | **0.057** | **0.181** | 0.221 | 0.382 | **0.068** | **0.199** |
| | 720 | 0.081 | 0.212 | **0.077** | **0.208** | **0.084** | **0.230** | 0.098 | 0.234 | 0.095 | 0.243 | **0.081** | **0.213** | 0.303 | 0.466 | **0.093** | **0.231** |
| ETTm2 | 96 | **0.063** | **0.183** | 0.063 | 0.186 | 0.097 | 0.244 | **0.060** | **0.183** | 0.128 | 0.278 | **0.068** | **0.195** | 0.073 | 0.200 | **0.069** | **0.193** |
| | 192 | **0.093** | **0.229** | **0.093** | 0.230 | 0.129 | 0.281 | **0.093** | **0.233** | 0.145 | 0.298 | **0.099** | **0.240** | 0.107 | 0.248 | **0.103** | **0.240** |
| | 336 | 0.120 | **0.263** | 0.119 | 0.264 | 0.174 | 0.326 | **0.129** | **0.276** | 0.148 | 0.303 | **0.123** | **0.269** | 0.163 | 0.314 | **0.135** | **0.281** |
| | 720 | 0.173 | **0.318** | **0.171** | 0.319 | 0.201 | 0.354 | **0.193** | **0.337** | 0.208 | 0.359 | **0.174** | **0.320** | 0.325 | 0.441 | **0.191** | **0.337** |

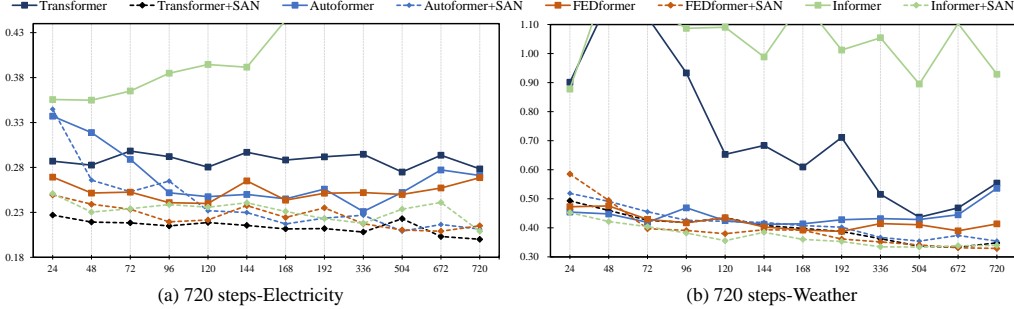

(a) 720 steps-Electricity

(b) 720 steps-Weather

Figure 4: The long-term forecasting MSE evaluations of different Transformer-based models under various input lengths. Large values are discarded to illustrate the overall trend better.

largely improved. When the input length is set to 720 on the Electricity dataset, the performance of Informer has been boosted by **77.83%** (from 0.9426 to 0.2090), and the average improvement on four backbones under the same setting is **52.55%**. Moreover, all of the backbones enhanced by SAN tend to produce more accurate forecasting as the length increases. To be specific, on the Weather dataset, Transformer achieves a reduction on MSE of **29.40%** when prolonging input from 24 steps to 720 steps, and the average improvement on four backbones is **33.11%**. These results greatly meet our expectations and also validate the effectiveness of SAN on various input lengths.

## B.4 Additional Prediction Showcases

We provide the additional comparison between SAN and other normalization methods in Fig. 5 with FEDformer [45] on various datasets. Clearly, SAN can better estimate the future distribution so as to help the backbone model to achieve superior performance, where the forecasting results are better aligned with the groundtruth.

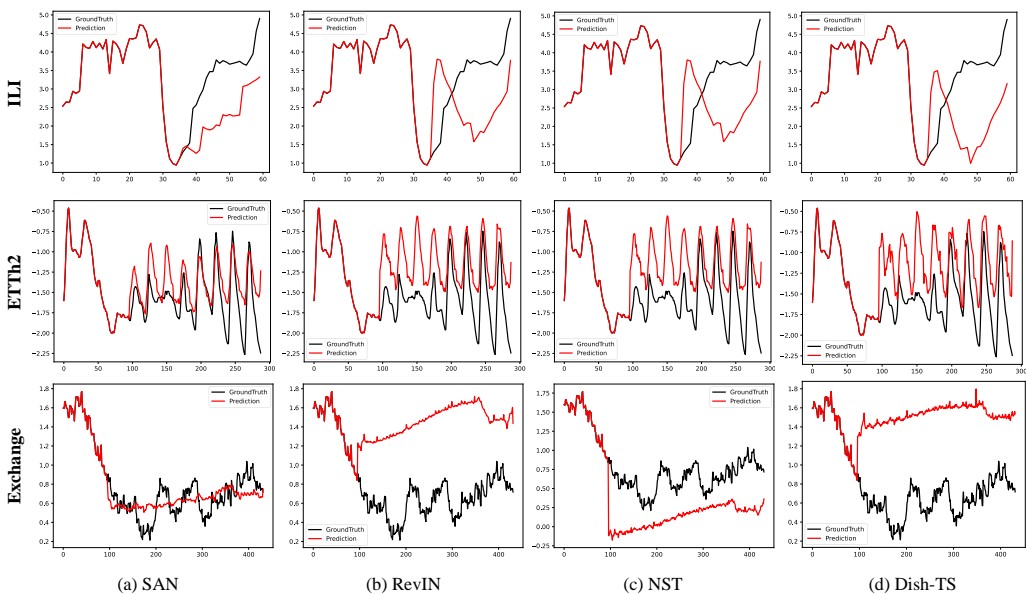

Figure 5: Illustration of the additional prediction showcases comparing SAN and baseline models. The experiment is conducted on the ILI, ETTh2, and Exchange dataset. Following the same input sequence length setting in our main experiments, the target sequence length is set to 24, 192, and 336 respectively.

## B.5 Ablation Study

**Statistic Prediction Module** In this section, we aim to analyze the effectiveness of our designs in the statistic prediction module. We instantiate our method and its variants on Autoformer and test their performance on two typical non-stationary datasets: Exchange and ETTh2. Similarly, we repeat the experiments three times with fixed seed and report the evaluations with standard deviation in Table 7.

Obviously, with the proposed two techniques combined, the statistic prediction module can achieve the best accuracy, leading to optimal forecasting performance. Besides, both *residual learning* and *individual preference* contribute positive effects and the former one is much more important, without which SAN can even bring negative effects to the backbone model. These results validate the rationality of our thoughts about the characteristics of the mean value and also reveal the importance of accurate modeling of future statistics to SAN. Besides, SAN without individual

Table 7: Forecasting errors under the multivariant setting with respect to variants of SAN. The best performance are highlighted in **bold**.

| Variants | SAN | | w/o individual | | w/o residual | | w/o SAN | |
|---|---|---|---|---|---|---|---|---|
| Metric | MSE | MAE | MSE | MAE | MSE | MAE | MSE | MAE |
| **Exchange** 96 | **0.082±0.001** | **0.208±0.001** | 0.089±0.005 | 0.209±0.007 | 0.135±0.003 | 0.264±0.001 | 0.152±0.006 | 0.283±0.007 |
| 192 | **0.157±0.001** | **0.296±0.003** | 0.184±0.009 | 0.306±0.010 | 0.331±0.044 | 0.416±0.025 | 0.369±0.055 | 0.437±0.033 |
| 336 | **0.262±0.004** | **0.385±0.002** | 0.340±0.001 | 0.422±0.001 | 0.658±0.044 | 0.593±0.024 | 0.534±0.130 | 0.544±0.066 |
| 720 | **0.689±0.043** | **0.629±0.020** | 0.982±0.001 | 0.753±0.002 | 1.456±0.011 | 0.882±0.009 | 1.222±0.099 | 0.848±0.021 |
| **ETTh2** 96 | **0.316±0.001** | **0.366±0.001** | 0.321±0.013 | 0.367±0.008 | 0.383±0.020 | 0.413±0.012 | 0.384±0.021 | 0.420±0.013 |
| 192 | **0.413±0.013** | 0.426±0.007 | 0.414±0.023 | **0.422±0.012** | 0.463±0.030 | 0.469±0.020 | 0.457±0.020 | 0.454±0.014 |
| 336 | **0.446±0.004** | 0.457±0.003 | 0.448±0.003 | **0.453±0.001** | 0.586±0.025 | 0.541±0.009 | 0.468±0.010 | 0.473±0.005 |
| 720 | **0.471±0.009** | **0.474±0.005** | 0.483±0.012 | 0.477±0.005 | 0.889±0.007 | 0.682±0.001 | 0.473±0.005 | 0.485±0.005 |

modeling performs well on the ETT2 dataset but performs poorly on the Exchange dataset. Such a phenomenon reveals that the evolving trends of different scenarios vary, and it is required to model the complex relationships among multiple variables individually. Moreover, since we only incorporate the properties of mean values into a simple MLP network, how to design a proper mechanism or network architecture for statistics modeling is a promising direction for optimizing our method, and we leave such explorations for future work.

**Slicing Length** The slicing length is a key parameter of SAN. We aim to study the effect of different slicing lengths on our method. Ablation experiments are conducted by using SCINet as the backbone model under the long-term forecasting setting ($L_{out} = 60$ for the ILI dataset and $L_{out} = 720$ for the rest datasets). Each experiment is conducted three times with a fixed random seed. The forecasting errors and the corresponding standard deviation are presented in Table 8.

Table 8: Forecasting errors under the multivariant setting with respect to different slicing lengths. The best performance are highlighted in **bold**.

| Slicing Length | 6 | | 12 | | 24 | | 48 | |
|---|---|---|---|---|---|---|---|---|
| Metric | MSE | MAE | MSE | MAE | MSE | MAE | MSE | MAE |
| Electricity | 0.210±0.002 | 0.305±0.002 | 0.207±0.002 | **0.305±0.002** | **0.206±0.004** | 0.307±0.003 | 0.208±0.002 | 0.307±0.001 |
| Exchange | **0.892±0.028** | **0.712±0.013** | 0.895±0.037 | **0.712±0.017** | 0.901±0.005 | 0.715±0.002 | 0.898±0.037 | 0.714±0.015 |
| Traffic | 0.612±0.001 | **0.376±0.001** | 0.608±0.002 | **0.373±0.001** | **0.607±0.001** | 0.381±0.001 | 0.611±0.002 | 0.382±0.002 |
| Weather | **0.338±0.002** | 0.366±0.002 | **0.338±0.001** | 0.365±0.002 | 0.340±0.001 | 0.367±0.001 | 0.339±0.001 | 0.366±0.001 |
| ILI | **2.487±0.034** | **1.063±0.008** | 2.680±0.055 | 1.118±0.015 | n/a | n/a | n/a | n/a |
| ETTh1 | 0.491±0.002 | 0.475±0.001 | **0.488±0.001** | 0.474±0.001 | 0.489±0.004 | **0.473±0.001** | 0.492±0.004 | 0.474±0.002 |
| ETTh2 | 0.440±0.001 | 0.465±0.001 | **0.435±0.002** | 0.460±0.002 | 0.437±0.007 | **0.459±0.006** | 0.443±0.007 | 0.462±0.004 |
| ETTm1 | 0.495±0.043 | 0.469±0.024 | **0.450±0.001** | **0.441±0.001** | 0.611±0.218 | 0.503±0.084 | 0.463±0.006 | 0.448±0.003 |
| ETTm2 | **0.391±0.001** | 0.406±0.001 | **0.391±0.001** | **0.405±0.001** | 0.392±0.001 | **0.405±0.001** | 0.403±0.009 | 0.415±0.006 |

Our heuristic selection of slicing length appears to be effective among the candidates, indicating that both artificially defined and actual periods are useful in selecting the optimal setting. Additionally, there were no significant performance differences observed under various settings, suggesting that SAN is resilient to changes in slicing length.

## B.6 Detailed Results of the Comparison between SAN and Normalization Methods

In Table 9, we provide the detailed experimental results of the comparison between SAN and state-of-the-art normalization methods for non-stationary time series forecasting: RevIN [17], NST [25] and Dish-TS [10]. We re-implement the former two methods and Dish-TS is implemented by its official code[11].

The table clearly shows that SAN outperforms existing approaches in most cases, except for the Weather dataset. Considering that the Weather dataset is the most stationary dataset, the results suggest that SAN can better remove the non-stationary factors in the raw data, even leading to an over-stationary issue that degrades the performance.

---

[11]https://github.com/weifantt/Dish-TS

Besides, Dish-TS performs poorly in the benchmark. While it addresses the distribution shift between input and horizon series, it fails to optimize both the coefficient network and backbone network for overlooking the intrinsic bi-level optimization target of distribution estimation and forecasting tasks. By adopting a joint training schema, Dish-TS disturbs both networks and results in poor performance in certain cases. On the opposite, SAN benefits from the proposed two-stage schema which decouples the two tasks. This allows for proper optimization of each component and leads to improved overall performance.

Table 9: Detailed results of the comparison between SAN and normalization methods. The best results are highlighted in **bold**.

| Methods | | FEDformer | | | | | | | | Autoformer | | | | | | | |
|---|---|---|---|---|---|---|---|---|---|---|---|---|---|---|---|---|---|
| | | + SAN | | + RevIN | | + NST | | +Dish-TS | | + SAN | | + RevIN | | + NST | | +Dish-TS | |
| Metric | | MSE | MAE | MSE | MAE | MSE | MAE | MSE | MAE | MSE | MAE | MSE | MAE | MSE | MAE | MSE | MAE |
| Electricity | 96 | **0.164** | **0.272** | 0.172 | 0.278 | 0.172 | 0.279 | 0.175 | 0.284 | **0.172** | **0.281** | 0.179 | 0.286 | 0.179 | 0.285 | 0.179 | 0.290 |
| | 192 | **0.179** | **0.286** | 0.185 | 0.289 | 0.187 | 0.291 | 0.188 | 0.296 | **0.195** | **0.300** | 0.216 | 0.316 | 0.209 | 0.309 | 0.215 | 0.318 |
| | 336 | **0.191** | **0.299** | 0.200 | 0.304 | 0.202 | 0.307 | 0.209 | 0.319 | **0.211** | **0.316** | 0.233 | 0.331 | 0.246 | 0.335 | 0.244 | 0.343 |
| | 720 | **0.230** | 0.334 | 0.243 | 0.337 | **0.230** | **0.326** | 0.239 | 0.343 | **0.236** | **0.335** | 0.246 | 0.341 | 0.252 | 0.345 | 0.286 | 0.370 |
| Exchange | 96 | **0.079** | **0.205** | 0.148 | 0.279 | 0.145 | 0.275 | 0.131 | 0.263 | **0.082** | **0.208** | 0.166 | 0.295 | 0.177 | 0.304 | 0.225 | 0.341 |
| | 192 | **0.156** | **0.295** | 0.266 | 0.377 | 0.274 | 0.383 | 0.538 | 0.523 | **0.157** | **0.296** | 0.299 | 0.404 | 0.275 | 0.385 | 0.760 | 0.610 |
| | 336 | **0.260** | **0.384** | 0.428 | 0.484 | 0.437 | 0.488 | 0.667 | 0.591 | **0.262** | **0.385** | 0.448 | 0.496 | 0.442 | 0.490 | 0.707 | 0.628 |
| | 720 | **0.697** | **0.633** | 1.056 | 0.789 | 1.064 | 0.787 | 1.480 | 0.954 | **0.689** | **0.629** | 1.068 | 0.791 | 1.049 | 0.784 | 2.341 | 1.063 |
| Traffic | 96 | **0.536** | **0.330** | 0.613 | 0.347 | 0.612 | 0.348 | 0.613 | 0.350 | **0.569** | **0.350** | 0.643 | 0.354 | 0.645 | 0.354 | 0.652 | 0.363 |
| | 192 | **0.565** | **0.345** | 0.637 | 0.356 | 0.641 | 0.357 | 0.644 | 0.362 | **0.594** | **0.364** | 0.659 | 0.373 | 0.643 | 0.367 | 0.669 | 0.374 |
| | 336 | **0.580** | **0.354** | 0.652 | 0.363 | 0.654 | 0.363 | 0.659 | 0.370 | **0.591** | **0.363** | 0.662 | 0.371 | 0.665 | **0.363** | 0.683 | 0.376 |
| | 720 | **0.607** | **0.367** | 0.686 | 0.382 | 0.688 | 0.380 | 0.693 | 0.388 | **0.623** | 0.380 | 0.700 | 0.384 | 0.667 | **0.373** | 0.703 | 0.392 |
| Weather | 96 | **0.179** | 0.239 | 0.187 | **0.234** | 0.187 | **0.234** | 0.244 | 0.317 | **0.194** | 0.256 | 0.212 | 0.257 | 0.211 | **0.254** | 0.268 | 0.338 |
| | 192 | 0.234 | 0.296 | **0.235** | **0.272** | **0.235** | **0.272** | 0.320 | 0.380 | **0.258** | 0.316 | 0.264 | **0.300** | 0.265 | 0.301 | 0.376 | 0.421 |
| | 336 | 0.304 | 0.348 | **0.287** | **0.307** | 0.289 | 0.308 | 0.424 | 0.452 | 0.329 | 0.367 | 0.309 | 0.329 | **0.303** | **0.324** | 0.476 | 0.486 |
| | 720 | 0.400 | 0.404 | 0.361 | 0.353 | **0.359** | **0.352** | 0.604 | 0.553 | 0.440 | 0.438 | 0.377 | 0.367 | **0.366** | **0.357** | 0.612 | 0.560 |
| ILI | 24 | **2.614** | **1.119** | 3.218 | 1.172 | 3.302 | 1.281 | 2.883 | 1.102 | **2.777** | **1.157** | 3.780 | 1.270 | 3.482 | 1.207 | 3.636 | 1.249 |
| | 36 | **2.537** | **1.079** | 3.055 | 1.135 | 3.193 | 1.240 | 2.865 | 1.077 | **2.649** | **1.104** | 3.114 | 1.157 | 3.423 | 1.289 | 3.284 | 1.178 |
| | 48 | **2.416** | **1.032** | 2.734 | 1.055 | 2.936 | 1.171 | 2.759 | 1.033 | **2.420** | **1.029** | 2.865 | 1.099 | 3.163 | 1.217 | 2.942 | 1.086 |
| | 60 | **2.299** | **1.003** | 2.841 | 1.095 | 2.904 | 1.173 | 2.878 | 1.075 | **2.401** | **1.021** | 2.846 | 1.104 | 2.871 | 1.140 | 2.856 | 1.083 |
| ETTh1 | 96 | **0.383** | **0.409** | 0.392 | 0.413 | 0.394 | 0.414 | 0.390 | 0.424 | 0.522 | 0.474 | 0.491 | 0.463 | 0.550 | 0.503 | **0.456** | **0.454** |
| | 192 | **0.431** | **0.438** | 0.443 | 0.444 | 0.441 | 0.442 | 0.441 | 0.458 | 0.498 | **0.472** | 0.513 | 0.478 | 0.530 | 0.492 | **0.495** | 0.480 |
| | 336 | **0.471** | **0.456** | 0.495 | 0.467 | 0.485 | 0.466 | 0.495 | 0.486 | 0.571 | 0.509 | 0.528 | 0.485 | **0.524** | **0.484** | 0.539 | 0.496 |
| | 720 | **0.504** | **0.488** | 0.520 | 0.498 | 0.505 | 0.496 | 0.519 | 0.509 | 0.555 | 0.514 | 0.543 | 0.510 | **0.510** | **0.491** | 0.563 | 0.522 |
| ETTh2 | 96 | **0.300** | **0.355** | 0.380 | 0.402 | 0.381 | 0.403 | 0.806 | 0.589 | **0.316** | **0.366** | 0.411 | 0.410 | 0.394 | 0.398 | 1.100 | 0.670 |
| | 192 | **0.392** | **0.413** | 0.457 | 0.443 | 0.478 | 0.453 | 0.936 | 0.659 | **0.413** | **0.426** | 0.478 | 0.450 | 0.473 | 0.450 | 0.976 | 0.672 |
| | 336 | **0.459** | **0.462** | 0.515 | 0.479 | 0.561 | 0.499 | 1.039 | 0.702 | **0.446** | **0.457** | 0.545 | 0.493 | 0.528 | 0.490 | 1.521 | 0.783 |
| | 720 | **0.462** | **0.472** | 0.507 | 0.487 | 0.502 | 0.481 | 1.237 | 0.759 | **0.471** | **0.474** | 0.523 | 0.490 | 0.524 | 0.498 | 1.105 | 0.745 |
| ETTm1 | 96 | **0.311** | **0.355** | 0.340 | 0.385 | 0.336 | 0.382 | 0.348 | 0.397 | **0.343** | **0.378** | 0.458 | 0.446 | 0.468 | 0.448 | 0.477 | 0.460 |
| | 192 | **0.351** | **0.383** | 0.390 | 0.411 | 0.386 | 0.409 | 0.406 | 0.428 | **0.390** | **0.400** | 0.560 | 0.491 | 0.526 | 0.468 | 0.545 | 0.488 |
| | 336 | **0.390** | **0.407** | 0.432 | 0.436 | 0.438 | 0.441 | 0.438 | 0.450 | **0.415** | **0.418** | 0.607 | 0.508 | 0.786 | 0.559 | 0.650 | 0.533 |
| | 720 | **0.456** | **0.444** | 0.497 | 0.466 | 0.483 | 0.460 | 0.497 | 0.481 | **0.476** | **0.453** | 0.623 | 0.526 | 0.564 | 0.501 | 0.595 | 0.518 |
| ETTm2 | 96 | **0.175** | **0.266** | 0.192 | 0.272 | 0.191 | 0.272 | 0.394 | 0.395 | 0.236 | 0.317 | **0.233** | **0.307** | 0.253 | 0.323 | 0.976 | 0.572 |
| | 192 | **0.246** | **0.315** | 0.270 | 0.320 | 0.270 | 0.321 | 0.552 | 0.472 | **0.260** | **0.329** | 0.288 | 0.337 | 0.289 | 0.335 | 0.532 | 0.485 |
| | 336 | **0.315** | **0.362** | 0.348 | 0.367 | 0.353 | 0.371 | 0.808 | 0.601 | **0.330** | 0.376 | 0.345 | 0.370 | 0.339 | **0.365** | 0.795 | 0.592 |
| | 720 | **0.412** | 0.422 | 0.430 | **0.415** | 0.445 | 0.422 | 1.282 | 0.771 | **0.417** | 0.428 | 0.434 | **0.419** | 0.426 | 0.432 | 1.271 | 0.768 |

## B.7  SAN for Slice-based Forecasting Methods

In this section, we turn to investigate the generalizability of SAN towards recently emerged slice-based forecasting methods: PatchTST [26] and Crossformer [41]. We build forecasting models using their official codes and hyper-parameter settings (if available) [12][13]. For PatchTST, we replace the RevIN[17] layer with our SAN. We report the experimental results on 5 datasets in Table 10.

The results demonstrate that SAN can improve the forecasting performance of both PatchTST and CrossFormer to some extent in most cases. The improvement for PatchTST is not significant due to two main reasons: 1) RevIN has already been introduced in the model to mitigate the impact of

---

[12]https://github.com/yuqinie98/PatchTST
[13]https://github.com/Thinklab-SJTU/Crossformer

Table 10: Forecasting accuracy of SAN applied to slice-based forecasters. The **bold** values indicate better performance.

| Methods Metric | | PatchTST MSE | MAE | + SAN MSE | MAE | Crossformer MSE | MAE | + SAN MSE | MAE |
|---|---|---|---|---|---|---|---|---|---|
| | 96 | 0.138 | **0.233** | **0.136** | 0.234 | 0.150 | 0.258 | **0.143** | **0.246** |
| | 192 | 0.153 | **0.247** | **0.150** | **0.247** | 0.175 | 0.284 | **0.162** | **0.265** |
| Electricity | 336 | 0.170 | **0.263** | **0.165** | 0.264 | 0.218 | 0.325 | **0.177** | **0.280** |
| | 720 | 0.206 | **0.296** | **0.200** | **0.296** | 0.226 | 0.324 | **0.221** | **0.318** |
| | 96 | 0.094 | **0.216** | **0.087** | 0.218 | 0.283 | 0.393 | **0.087** | **0.219** |
| | 192 | 0.191 | 0.311 | **0.181** | **0.323** | 1.087 | 0.804 | **0.171** | **0.313** |
| Exchange | 336 | 0.343 | 0.427 | **0.305** | **0.418** | 1.367 | 0.905 | **0.286** | **0.401** |
| | 720 | 0.888 | 0.706 | **0.659** | **0.620** | 1.546 | 0.987 | **0.749** | **0.653** |
| | 96 | **0.147** | **0.197** | 0.150 | 0.205 | **0.148** | 0.214 | 0.151 | **0.210** |
| | 192 | **0.191** | **0.240** | 0.194 | 0.252 | 0.201 | 0.270 | **0.198** | **0.253** |
| Weather | 336 | 0.244 | **0.282** | **0.243** | 0.290 | **0.248** | 0.311 | 0.248 | **0.294** |
| | 720 | 0.320 | **0.334** | **0.311** | 0.343 | 0.366 | 0.395 | **0.322** | **0.350** |
| | 96 | 0.382 | 0.403 | **0.375** | **0.398** | 0.390 | 0.417 | **0.387** | **0.402** |
| | 192 | 0.416 | 0.423 | **0.413** | **0.422** | 0.424 | 0.448 | **0.413** | **0.425** |
| ETTh1 | 336 | 0.441 | 0.440 | **0.428** | **0.434** | 0.486 | 0.492 | **0.436** | **0.431** |
| | 720 | 0.470 | 0.475 | **0.445** | **0.461** | 0.507 | 0.519 | **0.467** | **0.474** |
| | 96 | 0.174 | 0.261 | **0.167** | **0.260** | 0.330 | 0.401 | **0.170** | **0.262** |
| | 192 | 0.238 | 0.307 | **0.222** | **0.298** | 0.623 | 0.543 | **0.224** | **0.301** |
| ETTm2 | 336 | 0.293 | 0.346 | **0.276** | **0.334** | 0.887 | 0.637 | **0.274** | **0.333** |
| | 720 | 0.373 | 0.401 | **0.366** | **0.393** | 0.844 | 0.640 | **0.366** | **0.390** |

non-stationary time series; 2) Since both are slice-based methods that split series into slices (non-overlapping slices for SAN and overlapping patches for PatchTST), the parameter settings may have a greater impact on performance and how to determine proper settings require further explorations. Besides, The official code of CrossFormer provides parameter settings for Electricity, Weather, and ETTh1 datasets. For Exchange and ETTm2 datasets, we extracted common and reasonable settings. Without SAN, CrossFormer performs poorly compared to PatchTST on these latter two datasets due to unsuitable parameters. However, when enhanced with SAN under the same setting, CrossFormer achieves competitive or even superior performance. This phenomenon reveals that **SAN can potentially reduce reliance on parameter settings for backbone models while also reducing costs associated with parameter adjustment in real-world forecasting applications, further validating our attempt to divide complex non-stationary forecasting task into two easier subtasks through the two-stage training schema.**

## C Implementation Details

### C.1 Architecture of Statistic Prediction Module

The computation of $\text{MLP}(x_1, x_2)$ in our paper can be summarized as follows:

$$
\begin{aligned}
x_1 &= act_1(W_1 * x_1) \\
x_2 &= act_1(W_2 * x_2) \\
x &= [x_1; x_2] \\
output &= act_2(W_3 * x)
\end{aligned}
\tag{9}
$$

Here, the symbol [*;*] represents the concatenate operation. We set $act_1(), act_2() = Relu(), Relu()$ for standard deviation and the activate function of the mean is set to $Tanh(), Identity()$ respectively. $W_1, W_2, W_3$ are learnable transformation matrices with hidden sizes of {512,512,1024}.

### C.2 Algorithm of The Two-stage Training Schema

To apply SAN to backbone forecasting models, we propose a two-stage training schema to tackle the challenge of the bi-level optimization target. The statistics prediction module is first trained into

convergence, which is then frozen and treated as a plugin during the second stage of training the forecasting model. We provide the pseudo-code of such a procedure in Alg. 1.

---

**Algorithm 1** Two-stage Training Schema.

---

**Require:** Input series $X = \{\boldsymbol{x}^i\}_{i=1}^N$; Horizon series $Y = \{\boldsymbol{y}^i\}_{i=1}^N$; Slicing length $T$
  1: Initialize parameters $\phi, \theta$
  2: **while** not converge **do**
  3:     **for all** input $\boldsymbol{x}^i \in X$, horizon $\boldsymbol{y}^i \in Y$ **do**
  4:         Compute input statistics $\mu_j^i, \sigma_j^i$ by Eq. 1 with $T$
  5:         Predict future statistics $\hat{\boldsymbol{\mu}}^i, \hat{\boldsymbol{\sigma}}^i$ by Eq. 3 using $f_\phi(*)$
  6:         Update $\phi$ using loss function $l_{sp}$
  7:     **end for**
  8: **end while**                                          ▷ Training of the statistics prediction module
  9:
 10: **while** not converge **do**
 11:     **for all** input $\boldsymbol{x}^i \in X$, horizon $\boldsymbol{y}^i \in Y$ **do**
 12:         Compute input statistics $\mu_j^i, \sigma_j^i$ by Eq. 1 with $T$
 13:         Normalize input series to $\bar{\boldsymbol{x}}^i$ by Eq. 2
 14:         Forecast $\bar{\boldsymbol{y}}^i = g_\theta(\bar{\boldsymbol{x}}^i)$
 15:         Predict future statistics $\hat{\boldsymbol{\mu}}^i, \hat{\boldsymbol{\sigma}}^i$ by Eq. 3 using $f_\phi(*)$
 16:         $\hat{\boldsymbol{\mu}}^i$.detach(), $\hat{\boldsymbol{\sigma}}^i$.detach()         ▷ Stop-gradient, freeze the statistics prediction module
 17:         Denormalize $\bar{\boldsymbol{y}}^i$ to $\hat{\boldsymbol{y}}^i$ by Eq. 4
 18:         Update $\theta$ using loss function $l_{fc}$
 19:     **end for**
 20: **end while**                                                      ▷ Training of the forecasting model

---

# D   Limitations

Though SAN shows promising performance on the benchmark dataset, there are still some limitations of this method. First is that we mainly select the slicing length heuristically or search in predefined candidates and the current design cannot handle indivisible length or the multi-period characteristic of time series. Such a solution works for the experiments but lacks generality in real-world applications. Second is that SAN may lead to an over-stationary issue, leading to sub-optimal performance. Moreover, when applied to similar slice-based methods (especially overlapping-slice-based ones), determining how to adjust SAN's parameters for them is not a straightforward task. Therefore, a more flexible solution with automatic slicing length selection and normalization intensity control will be our exploring direction.

