# OpenReview forum: "Adaptive Normalization for Non-stationary Time Series Forecasting: A Temporal Slice Perspective"
_NeurIPS.cc/2023/Conference — NeurIPS 2023 poster_

### Official Review · Reviewer_3LTc · 2023-06-16

**Soundness:** 3 good
**Presentation:** 2 fair
**Contribution:** 3 good
**Rating:** 6
**Confidence:** 3

**Summary:**

The paper proposes a normalization technique that works on sliced time-series with the main goal to remove non-stationary behavior of the inputs (and outputs). The paper computes mean and standard deviation of the slides inputs and then normalizes the inputs by them. Additionally, the paper proposes to estimate output normalization for y from the input mean and standard devation.

**Strengths:**

Problem statement is well presented.

**Weaknesses:**

The paper proposes to normalize the slice of a time series by its mean and standard deviation of its inputs.

There is a critical issue with that: this is a **non-causal** normalization, i.e., inputs at the start of the time series are normalized by values observed later in the slice. This makes the proposed approach **unsuitable for forecasting**. It is very likely that the good empirical results presented in the paper are due to this leakage of future values of the inputs **x**.

Secondly, assuming that the first problem can be fixed, there are already very similar works presented in the literature. The authors do  mention them but do not compare. Given the practical nature of the paper, these other approaches should be compared as the difference seems quite minor from a general theoretical view point. For example, DAIN (https://arxiv.org/pdf/1902.07892.pdf) is very similar and uses normalization by mean and standard deviations, mixed with neural networks. Please explain why the proposed method is novel compared to DAIN and provide experimental support that it actually improves upon DAIN.

**Questions:**

* Why should the proposed method perform better than DAIN, which also uses similar normalization ideas (mean and stdev).
* What is the variable that equation 1 is summing over?

**Limitations:**

No limitations were mentioned in the paper. But clearly the choice of the length of the slicing window is a problem for the proposed method, compared to other methods. The authors should either clearly state that this is a limitation or make experiments showing that the method performance is not affected by the slice length.

---

> ### Author Rebuttal · Authors · 2023-08-09
>
> Thank you for your acknowledgement and valuable feedbacks on our work, we would like to address your concerns as follows.
>
> - In practice, **it is commonly assumed that time series data within a slice have the same distribution**. Existing normalization methods (such as DAIN and RevIN) also assume that the entire input/output series follows the same distribution. Therefore, conducting non-causal normalization is reasonable. Additionally, our statistics prediction module and backbone model are **trained solely on the training set**. In the testing phase, the mean and standard deviations for output series are **predicted based on the input series**, ensuring no issue of information leakage.
>
> - DAIN is a pioneering work that introduces normalization into forecasting tasks. However, there are two main drawbacks of DAIN compared to our proposed SAN:
>
>   - Firstly, DAIN does not **restore the non-stationary information back** to the output of forecasting models. For example, given two simulated series $x_1= 0.1sin(t)$, $x_2 = 100sin(t)$, DAIN may normalize them into similar input series $\bar{x_1}\approx\bar{x_2}\approx sin(t)$ for later processing. This can lead to the forecasting model predicting similar outputs that greatly violate the original scale. On the other hand, SAN effectively addresses this challenge by **learning to predict future distribution and performing de-normalization operations with predicted statistics**.
>   - Secondly, DAIN roughly assumes that the **entire input series follows the same distribution**. As illustrated in Figure 1 in our paper, it is clear that distribution changes occur within small slices and there exists a discrepancy in distribution throughout the whole series. To tackle this issue well, SAN proposes slice-level adaptive normalization.
>
>   In summary, SAN is theoretically superior to DAIN. To further validate our claim, we conducted additional experiments comparing SAN and DAIN using DLinear on 4 datasets. The detailed results are provided in the following table which experimentally confirms the effectiveness of our method compared to DAIN.
>
>   |                 |                 | DLinear_SAN |           | DLinear_DAIN |         |
>   | --------------- | --------------- | ----------- | --------- | ------------ | ------- |
>   | **Dataset**     | **Pred_length** | **MSE**     | **MAE**   | **MSE**      | **MAE** |
>   | **Electricity** | 96              | **0.137**   | **0.234** | 0.203        | 0.314   |
>   |                 | 192             | **0.151**   | **0.247** | 0.217        | 0.328   |
>   |                 | 336             | **0.166**   | **0.264** | 0.229        | 0.339   |
>   |                 | 720             | **0.201**   | **0.295** | 0.248        | 0.352   |
>   | **Exchange**    | 96              | **0.085**   | **0.214** | 1.289        | 0.916   |
>   |                 | 192             | **0.177**   | **0.317** | 1.503        | 0.998   |
>   |                 | 336             | **0.294**   | **0.407** | 1.857        | 1.106   |
>   |                 | 720             | **0.726**   | **0.649** | 2.379        | 1.236   |
>   | **Weather**     | 96              | **0.152**   | **0.210** | 0.248        | 0.331   |
>   |                 | 192             | **0.196**   | **0.254** | 0.286        | 0.357   |
>   |                 | 336             | **0.246**   | **0.294** | 0.330        | 0.382   |
>   |                 | 720             | **0.315**   | **0.346** | 0.398        | 0.425   |
>   | **ETTh1**       | 96              | **0.383**   | **0.399** | 0.701        | 0.647   |
>   |                 | 192             | **0.419**   | **0.419** | 0.725        | 0.653   |
>   |                 | 336             | **0.437**   | **0.432** | 0.743        | 0.661   |
>   |                 | 720             | **0.446**   | **0.459** | 0.742        | 0.670   |
>
> - Thank you for bringing the writing flaw in equation 1 to our attention. The variable being summed is time points. We will carefully review and rectify any similar errors to enhance the clarity of our presentation.
>
> - Due to page limitations, we had discussed the limitations in section 4 of our supplementary material. We will provide more detailed discussions on the limitations of our method in a later version of our paper to better illustrate potential drawbacks. Additionally, we had studied the effect of slicing length in our supplementary material. Using SCINet as the backbone model, here are the corresponding results. It is evident that our proposed **SAN remains resilient to changes in slicing length**.
>
>   | slicing length | 6         |           | 12        |           | 24        |           | 48    |       |
>   | -------------- | --------- | --------- | --------- | --------- | --------- | --------- | ----- | ----- |
>   |                | MSE       | MAE       | MSE       | MAE       | MSE       | MAE       | MSE   | MAE   |
>   | Electricity    | 0.210     | **0.305** | 0.207     | **0.305** | **0.206** | 0.307     | 0.208 | 0.307 |
>   | Exchange       | **0.892** | **0.712** | 0.895     | **0.712** | 0.901     | 0.715     | 0.898 | 0.714 |
>   | Traffic        | 0.612     | **0.376** | 0.608     | **0.373** | **0.607** | 0.381     | 0.611 | 0.382 |
>   | Weather        | **0.338** | 0.366     | **0.338** | **0.365** | 0.340     | 0.367     | 0.339 | 0.366 |
>   | ILI            | **2.487** | **1.063** | 2.680     | 1.118     | n/a       | n/a       | n/a   | n/a   |
>   | ETTh1          | 0.491     | 0.475     | **0.488** | 0.474     | 0.489     | **0.473** | 0.492 | 0.474 |
>   | ETTh2          | 0.440     | 0.465     | **0.435** | 0.460     | 0.437     | **0.459** | 0.443 | 0.462 |
>   | ETTm1          | 0.495     | 0.469     | **0.450** | **0.441** | 0.611     | 0.503     | 0.463 | 0.448 |
>   | ETTm2          | **0.391** | 0.406     | **0.391** | **0.405** | 0.392     | **0.405** | 0.403 | 0.415 |
>
> We hope that our responses have adequately addressed your concerns, and we appreciate the opportunity to clarify our work.

---

> > ### Comment · Reviewer_3LTc · 2023-08-19
> > **I keep my original scores unchanged**
> >
> > I would like to thank authors for their comments.
> >
> > * Unfortunately, the key issue of **non-causality** of the method, is still not addressed. The non-causality here means that the method leaks future values into the final forecasts, both during training and during **validation** (**testing**).
> > * The authors claim that it is not an issue because the method uses forecasting during inference. However, the **Equation 3**, which specifies how $\hat\mu^i$ and $\hat\sigma^i$ are estimated, is also **non-causal**, i.e., the MLPs take as inputs the means and stdevs of **all slices** of the i-th input series ($\mu^i$ and $\sigma^i$). For example, to calculate the mean of the first slice ($\hat\mu^1$) one needs to have access to the means ($\mu^i_1$, $\mu^i_2$, ...) of **every future slice in i-th the series**.
> >
> > Therefore, I will keep my original score as the method as currently presented clearly leaks future values into the forecasts of $\hat\mu^i$ and $\hat\sigma^i$ and consequently to the the final outputs of the models.

---

> > > ### Author Response · Authors · 2023-08-20
> > > **Response to Reviewer 3LTc**
> > >
> > > Thank you for your response. We would like to further elaborate on the functionality of Equation 3 and address any concerns regarding potential information leakage:
> > >
> > > - Firstly, in the forecasting tasks, **the input series and target series is not overlapped**.
> > > - Secondly, for future statistics $\hat{\mu}, \hat{\sigma}$, they are **estimated only by the input/observed series' statistics** $\mu, \sigma$.
> > > - As pointed out by the reviewer, the calculation of the mean $\hat{\mu_i}$ for the first target slice indeed requires accessing the means $\mu_1, \mu_2, \ldots$ associated with various slices. However, **it is crucial to note that these $\mu_i$ values pertain to input/observed means, rather than future means.**
> > >
> > > For example, under a setting of input-96-predict-192 with a slice length of 12, we can get non-overlapped input series $x_1$ and the corresponding target series $y_i$, where $|x_i| =96, |y_i|=192$. To estimate the means and stdevs of $y_i$'s 16 slices, our model utilizes the statistics of $x_i$'s 8 slices with the Equation 3. This explicit demonstration underscores that our modeling approach does not inadvertently expose any future information.
> > >
> > > We sincerely hope that this clarification addresses your concerns. Should you have any further inquiries or reservations, please do not hesitate to share them with us.

---

> > > > ### Comment · Reviewer_3LTc · 2023-08-20
> > > >
> > > > I appreciate your prompt response.
> > > >
> > > > Your comments resolve my worries about the method. Please also clarify the mathematical notation in the paper (Equation 3) so readers are can have a quick correct understanding of the method.
> > > >
> > > > Therefore, I have increased my score.

---

> ### Comment · Area_Chair_bXno · 2023-08-19
> **Request for Reviewer 3LTc to respond to the rebuttal**
>
> Reviewer 3LTc, as there are only 2 days left in the author discussion period, would you please read the authors' response, explain the extent to which their answers address your concerns, and whether you will adjust your rating.
>
> If you decide to keep your score, please justify this decision, specifying which aspects of the paper or response have been the deciding factors in you keeping your score.

---

### Official Review · Reviewer_jRLw · 2023-06-21

**Soundness:** 3 good
**Presentation:** 3 good
**Contribution:** 3 good
**Rating:** 6
**Confidence:** 4

**Summary:**

The paper introduces a novel approach called Slicing Adaptive Normalization (SAN) for non-stationary time series forecasting. The proposed method addresses the challenge of accurate predictions in the presence of non-stationarity in real-world data. It overcomes limitations in existing normalization techniques by considering the distribution discrepancy between input and horizon series and by modeling the evolving trends of statistical properties at a fine-grained temporal slice level. SAN is a model-agnostic framework that can be applied to various forecasting models, and experiments on benchmark datasets demonstrate its effectiveness.

**Strengths:**

Addressing non-stationarity: The paper tackles the problem of non-stationarity in time series forecasting, which is a significant challenge in real-world scenarios. By considering the distribution discrepancy between input and horizon series, SAN provides a mechanism to mitigate the impact of non-stationary nature on predictions.

Fine-grained normalization: SAN introduces a slice-level adaptive normalization approach, which operates on local temporal slices (sub-series) rather than the entire instance. This fine-grained normalization allows for a more accurate representation of the statistical properties within each slice, preserving distinct patterns and avoiding suboptimal improvements.

Evolving trends modeling: SAN incorporates a statistics prediction module to independently model the evolving trends of statistical properties in raw time series. This module improves the estimation of future distributions, enabling adaptive denormalization and enhancing the accuracy of predictions.

Model-agnostic framework: SAN is designed to be a general framework that can be applied to arbitrary forecasting models. It can serve as a plugin to existing models, making it flexible and adaptable to different forecasting scenarios.

**Weaknesses:**

The overall method is simple and easy to understand, however, I have the following concerns or questions.

[Major] Experiments. Most of my concerns are from the experimental section. (1) The baselines are not sufficient. The advanced models such as PatchTST [1], and crossformer[2] are not compared. If the method can also improve the advanced backbones, I will increase my rating.

[1] Nie, Yuqi, et al. "A Time Series is Worth 64 Words: Long-term Forecasting with Transformers." arXiv preprint arXiv:2211.14730 (2022).
[2] Zhang, Yunhao, and Junchi Yan. "Crossformer: Transformer utilizing cross-dimension dependency for multivariate time series forecasting." The Eleventh International Conference on Learning Representations. 2023.

Limited discussion of limitations: The paper does not extensively discuss the limitations or potential challenges of the proposed method. Providing a more thorough analysis of the limitations and potential drawbacks would give a clearer understanding of the scope and applicability of SAN.



**Questions:**

see Weaknesses

**Limitations:**

see Weaknesses

---

> ### Author Rebuttal · Authors · 2023-08-09
>
> We greatly appreciate your recognition of our proposal's novelty and effectiveness. We would like to address your concerns as follows:
>
> - Firstly, we conducted additional experiments using PatchTST and CrossFormer on 5 datasets. We built forecasting models using their official codes and hyper-parameter settings (if available). For PatchTST, we replaced the RevIN layer with our SAN. The detailed results and conclusions are provided in the table below.
>
>   |                 |                 | PatchTST  |           | PatchTST_SAN |           | CrossFormer |         | CrossFormer_SAN |           |
>   | --------------- | --------------- | --------- | --------- | ------------ | --------- | ----------- | ------- | --------------- | --------- |
>   | **Dataset**     | **Pred_length** | **MSE**   | **MAE**   | **MSE**      | **MAE**   | **MSE**     | **MAE** | **MSE**         | **MAE**   |
>   | **Electricity** | 96              | 0.138     | **0.233** | **0.136**    | 0.234     | 0.150       | 0.258   | **0.143**       | **0.246** |
>   |                 | 192             | 0.153     | **0.247** | **0.150**    | **0.247** | 0.175       | 0.284   | **0.162**       | **0.265** |
>   |                 | 336             | 0.169     | **0.263** | **0.165**    | 0.264     | 0.218       | 0.325   | **0.177**       | **0.280** |
>   |                 | 720             | 0.208     | **0.296** | **0.200**    | **0.296** | 0.226       | 0.324   | **0.221**       | **0.318** |
>   | **Exchange**    | 96              | 0.089     | **0.210** | **0.088**    | 0.221     | 0.283       | 0.393   | **0.087**       | **0.219** |
>   |                 | 192             | 0.195     | 0.315     | **0.174**    | **0.314** | 1.087       | 0.804   | **0.171**       | **0.313** |
>   |                 | 336             | 0.354     | 0.434     | **0.310**    | **0.421** | 1.367       | 0.905   | **0.286**       | **0.401** |
>   |                 | 720             | 0.869     | 0.698     | **0.705**    | **0.635** | 1.546       | 0.987   | **0.749**       | **0.653** |
>   | **Weather**     | 96              | 0.155     | **0.205** | **0.150**    | **0.205** | **0.148**   | 0.214   | 0.151           | **0.210** |
>   |                 | 192             | 0.200     | **0.245** | **0.195**    | 0.250     | 0.201       | 0.270   | **0.198**       | **0.253** |
>   |                 | 336             | 0.251     | **0.286** | **0.245**    | 0.290     | **0.248**   | 0.311   | **0.248**       | **0.294** |
>   |                 | 720             | 0.320     | **0.336** | **0.313**    | 0.340     | 0.366       | 0.395   | **0.322**       | **0.350** |
>   | **ETTh1**       | 96              | 0.408     | 0.425     | **0.378**    | **0.401** | 0.390       | 0.417   | **0.387**       | **0.402** |
>   |                 | 192             | 0.438     | 0.441     | **0.416**    | **0.424** | 0.424       | 0.448   | **0.413**       | **0.425** |
>   |                 | 336             | 0.442     | 0.446     | **0.428**    | **0.434** | 0.486       | 0.492   | **0.436**       | **0.431** |
>   |                 | 720             | 0.453     | 0.470     | **0.445**    | **0.461** | 0.507       | 0.519   | **0.467**       | **0.474** |
>   | **ETTm2**       | 96              | 0.168     | **0.257** | **0.166**    | 0.258     | 0.330       | 0.401   | **0.170**       | **0.262** |
>   |                 | 192             | 0.223     | **0.295** | **0.222**    | 0.302     | 0.623       | 0.543   | **0.224**       | **0.301** |
>   |                 | 336             | **0.280** | **0.335** | 0.302        | 0.353     | 0.887       | 0.637   | **0.274**       | **0.333** |
>   |                 | 720             | **0.371** | **0.391** | 0.402        | 0.418     | 0.844       | 0.640   | **0.366**       | **0.390** |
>
>   - SAN can improve the forecasting performance of both PatchTST and CrossFormer to some extent in most cases.
>   - The improvement for PatchTST is not significant due to two main reasons: 1) RevIN has already been introduced in the model to mitigate the impact of non-stationary time series; 2) We have not tuned the hyper-parameters of SAN combined with PatchTST, these results may not reflect its near-optimal performance. Since both are patch-based methods that split series into slices (non-overlapping slices for SAN and overlapping patches for PatchTST), parameter settings may have a greater impact on performance compared to non-patch-based models. These preliminary untuned experiments demonstrate the potential application of SAN in advanced methods.
>   - The official code of CrossFormer provides parameter settings for Electricity, Weather, and ETTh1 datasets. For Exchange and ETTm2 datasets, we extracted common and reasonable settings. Without SAN, CrossFormer performs poorly compared to PatchTST on these latter two datasets due to unsuitable parameters. However, when enhanced with SAN under the same setting, CrossFormer achieves competitive or even superior performance. This phenomenon reveals that   SAN can potentially **reduce reliance on parameter settings for backbone models while also reduce costs associated with parameter adjustment in real-world forecasting applications**.
>
> - Secondly, we would provide more in-detail discussions on the limitation of our method in the later version of our paper. This will help to better illustrate potential drawbacks based on the current discussions in Section 4 of our supplementary material. In short, SAN has three main limitations: One is that the current non-overlapping isometric slicing scheme is not flexible enough. Additionally, SAN may result in an over-stationary issue that negatively impacts performance. Lastly, as mentioned earlier, determining how to adjust SAN's parameters for patch-based models is not a straightforward task.
>
> We hope that our responses have adequately addressed your concerns, and we appreciate the opportunity to clarify our work.

---

> > ### Comment · Reviewer_jRLw · 2023-08-16
> > **Official Comment by reviewer jRLw**
> >
> > I appreciate your prompt response and the additional experiments you have conducted. While the new experiments provide valuable insights into the performance of your proposed method, I will be retaining my original score for your paper.
> >
> > I believe that further improvements can be made to enhance the clarity and depth of your article from two specific aspects:
> >
> > 1. Consider conducting additional analysis, such as delving into data patterns or engaging in statistical analysis, to elucidate the reasons behind the observed performance disparities between SAN and ReVIN when integrated with PatchTST. This could provide valuable insights into the specific scenarios where each method excels and help to uncover potential nuances that contribute to the contrasting results.
> >
> > 2. Address the variance in improvement magnitudes across different cases. Offering comprehensive explanations for why improvements are marginal in certain instances and significant in others would provide a more nuanced understanding for readers. By delineating the specific conditions or characteristics under which your proposed normalization method proves superior, you can offer valuable guidance to practitioners and researchers alike.
> >
> > These suggested enhancements would not only enrich the overall quality of your article but also contribute to a more comprehensive understanding of the strengths and limitations of your approach.

---

### Official Review · Reviewer_uWXn · 2023-07-05

**Soundness:** 2 fair
**Presentation:** 2 fair
**Contribution:** 2 fair
**Rating:** 5
**Confidence:** 2

**Summary:**

  Non-stationary time series forecasting is a challenging problem, and recent research has focused on using normalization techniques to address non-stationarity. However, these methods have limitations when it comes to handling the distribution discrepancy between the input and the forecasted horizon. This discrepancy arises from the assumption that all time points within the same instance share the same statistical properties. To overcome this limitation, the authors propose a method called SAN (sliced-level adaptive normalization) that empowers time series forecasting with more flexible normalization and denormalization.
SAN addresses this issue in several ways. First, it utilizes local temporal slices instead of considering the entire global instance, allowing for more localized analysis and adaptation. Second, SAN incorporates a slight network module that independently models the evolving trends of the statistical properties of the raw time series. This approach enables the model to adapt to changing statistical characteristics over time. Finally, SAN is a general model-agnostic plugin, which means it can be integrated into various existing forecasting models to enhance their performance.


**Strengths:**

- The authors emphasize the importance of adopting a local perspective for adaptive normalization. By considering local temporal slices instead of the entire global instance, the model gains a more granular understanding of the data, allowing for better adaptation to each slice's specific characteristics and trends. This localized approach enables more accurate and flexible normalization, leading to improved forecasting performance.
- The experimental results presented in the paper demonstrate promising performance in time series forecasting. By incorporating the SAN (sliced-level adaptive normalization) technique into existing forecasting models, the authors achieve notable improvements in mse/mae and predictive capabilities.


**Weaknesses:**

A. Insufficient justification for the problem definition.
    i. The main contribution of this paper is to perform normalization based on locally slicing the time series data, instead of assuming the entire time series follows the same distribution.
    ii. However, there is a lack of explanation regarding why global normalization should not be used. It would be beneficial to provide a detailed explanation, for example, by referencing papers on CPD (change point detection) or OOD (Out-of-Distribution) analysis within the same time series data, to clearly define the reasons for adopting a local perspective.
B.  Citation in Section 2.1. – Detailed weakness and gentle recommendation
    i. In section 2.1, discussing the RNN sentence, it would be better to cite papers on RNN, LSTM, and GRU to provide more comprehensive coverage (96-97).
    ii. Furthermore, in the following sentence, while Informer is cited for pointing out issues with RNN, it might be more suitable in the context to cite papers addressing RNN’s gradient exploding or vanishing problems.
    iii. In the subsequent sentence, when discussing self-attention and convolution for time series forecasting, it would be more appropriate to cite Scinet alongside Informer.
    iv. Following that, the mention of Fedformer and Autoformer, which incorporate decomposition characteristics from Transformer-based architecture, could be supplemented by referencing Dlinear.
C. Section 2.2 contents
    i. Section 2.2 introduces existing methodologies that employ adaptive normalization for non-stationary time series forecasting.
    ii. However, there is a lack of discussion on the limitations of previous research and the rationale behind the need for this study.
D. Section 3 – the methodology for slicing
    i.	The methodology of slicing the data, which is the most crucial aspect of this study, is not well understood.
    ii.	If equally sized segments were used, how were the discrepancies in length handled at the end of the slices?
    iii.	Additionally, it seems fitting to explore the application of methods such as DTW or CPD for segmentation based on similarity or probability in this study. Were they considered and applied?
E. The data example of Figure 2
    i.	The illustration in Figure 2 appears to represent variations in the magnitude of a sine wave, which does not seem suitable as an example of non-stationary data.
    ii.	It is suggested to replace the illustration in Figure 2 with a more non-stationary dataset and provide a detailed diagram depicting the slicing process and the resulting changes after normalization.
F.  Experiments
    i.	How do authors perceive the increase in learning complexity and inefficiency due to SAN being trained separately from the forecasting model as a module?
    ii.	The backbone models used in the experiments leverage decomposition and excel in handling stationary data. Is there a specific reason for applying SAN, designed for non-stationary learning, to these models?
    iii.	PatchTST also involves patching time series sub-sequences and applying them to a transformer-based architecture. Was SAN also applied to PatchTST?


**Questions:**

Please address my concerns in the above weakness

**Limitations:**

Adaptive normalization is one of the commonly used methodologies for handling non-stationary time series data. The application of slicing in this context is a contribution of this paper. However, the motivation behind this approach is not clearly explained.

---

> ### Author Rebuttal · Authors · 2023-08-09
>
> We greatly appreciate your acknowledgement of our proposal. We would like to address your concerns as follows:
>
> - Weakness A: Thank you for your valuable advice on defining the problem. We have found it helpful to reference papers on CPD or OOD to better illustrate the limitations of existing normalization methods and our motivation for proposing the slicing approach.
>
> - Weakness B&C: We appreciate your suggestions on describing limitations of existing methods. We will provide the corresponding discussions to better illustrate our motivation and contribution in this section. Additionally, we will improve our citations to provide a clearer background on forecasting.
>
> - Weakness D:
>
>   -  i. The idea of slicing time series is based on the observation that many real-world time series data exhibit sliced distribution shift. For example, in Figure 1 of our paper, the energy consumption data's scale changes daily while the approximate curve without absolute scale remains near constant throughout the day. Therefore, we aim to model this nature through the slicing operation.
>   -  ii. The current design of SAN cannot handle indivisible series lengths, so in our experiments, we enforce a slicing length that satisfies this condition. However, this non-overlapping isometric slicing scheme is not flexible and is one of the main limitations of our method as described in Section 4 of the supplementary material.
>   -  iii. This is an excellent question. We are currently exploring incorporating CPD methods to achieve a non-isometric slicing schema for increased flexibility and better performance.
>
> - Weakness E: Thank you for your advice regarding the framework diagram. We will consider your suggestion and redraw a more suitable diagram by replacing the simulated data.
>
> - Weakness F:
>
>   - i. Our statistics prediction module is a lightweight network consisting of only 2 MLPs, resulting in minimal training overhead in terms of time and memory. For instance, when training Autoformer enhanced with SAN on the ETTh1 dataset (input length 96, output length 720), the statistics prediction module takes only about 5 seconds to train, while Autoformer consumes approximately 35 seconds per epoch. Besides, thanks to the two-stage training approach, SAN introduces almost no additional delay in training backbone models.
>
>   - ii. Although many backbones used in our experiments employ decomposition techniques, these methods do not claim to be specifically designed for forecasting stationary data. The purpose of using these models as backbones is to test the generalization ability of SAN when applied to different architectures.
>
>   - iii. SAN can also be applied to PatchTST by replacing its RevIN layer. We provide comparison experiments as follows:
>
>     |                 |                 | PatchTST  |           | PatchTST_SAN |           |
>     | --------------- | --------------- | --------- | --------- | ------------ | --------- |
>     | **Dataset**     | **Pred_length** | **MSE**   | **MAE**   | **MSE**      | **MAE**   |
>     | **Electricity** | 96              | 0.138     | **0.233** | **0.136**    | 0.234     |
>     |                 | 192             | 0.153     | **0.247** | **0.150**    | **0.247** |
>     |                 | 336             | 0.169     | **0.263** | **0.165**    | 0.264     |
>     |                 | 720             | 0.208     | **0.296** | **0.200**    | **0.296** |
>     | **Exchange**    | 96              | 0.089     | **0.210** | **0.088**    | 0.221     |
>     |                 | 192             | 0.195     | 0.315     | **0.174**    | **0.314** |
>     |                 | 336             | 0.354     | 0.434     | **0.310**    | **0.421** |
>     |                 | 720             | 0.869     | 0.698     | **0.705**    | **0.635** |
>     | **Weather**     | 96              | 0.155     | **0.205** | **0.150**    | **0.205** |
>     |                 | 192             | 0.200     | **0.245** | **0.195**    | 0.250     |
>     |                 | 336             | 0.251     | **0.286** | **0.245**    | 0.290     |
>     |                 | 720             | 0.320     | **0.336** | **0.313**    | 0.340     |
>     | **ETTh1**       | 96              | 0.408     | 0.425     | **0.378**    | **0.401** |
>     |                 | 192             | 0.438     | 0.441     | **0.416**    | **0.424** |
>     |                 | 336             | 0.442     | 0.446     | **0.428**    | **0.434** |
>     |                 | 720             | 0.453     | 0.470     | **0.445**    | **0.461** |
>     | **ETTm2**       | 96              | 0.168     | **0.257** | **0.166**    | 0.258     |
>     |                 | 192             | 0.223     | **0.295** | **0.222**    | 0.302     |
>     |                 | 336             | **0.280** | **0.335** | 0.302        | 0.353     |
>     |                 | 720             | **0.371** | **0.391** | 0.402        | 0.418     |
>
>     It reveals that SAN can also boost the performance of PatchTST in most cases. While the improvement is not significant, we address it to two reasons: firstly, PatchTST already incorporates RevIN to mitigate non-stationarity and performs close to state-of-the-art compared to other forecasting approaches in most scenarios; secondly, we have not fine-tuned the hyperparameters for combining SAN with PatchTST yet, so the current results may not reflect optimal performance. As both SAN and PatchTST are patch-based methods that divide time series into segments (non-overlapping slices for SAN and overlapping patches for PatchTST), parameter settings may have a greater impact on their performance compared to non-patch-based models. These preliminary untuned experiments confirm the potential applicability of SAN in advanced methods.
>
> We hope that our responses have adequately addressed your concerns, and we appreciate the opportunity to clarify our work.

---

> > ### Comment · Reviewer_uWXn · 2023-08-18
> >
> > Thank you for your insightful rebuttal, and it addresses many of my concerns. However, there are still a few remaining concerns left and I prefer to stick to my previous score (Borderline accept).
> > There are many things to be modified in original manuscripts. For examples, Experiment sections are not yet organized well. (e.g. ablation studies of SAN and comparing or applying to state-of-the-art models)

---

### Official Review · Reviewer_roaz · 2023-07-25

**Soundness:** 2 fair
**Presentation:** 3 good
**Contribution:** 2 fair
**Rating:** 6
**Confidence:** 4

**Summary:**

The paper introduces a normalization approach for predicting non-stationary timeseries. While previous work on this topic assumes that output timeseries roughly share the same statistics (mean and variance) as the input timeseries and adjusts predictions accordingly, the authors propose two adjustments. Firstly, they predict the statistics of the target timeseries instead of relying solely on those of the input. Secondly, they utilize local time statistics from slices of the timeseries data, rather than global statistics from the entire timeseries. This approach outperforms alternative methods numerically.

**Strengths:**

- The proposed approach, SAN, is model-agnostic and can be applied to any timeseries regression model.
- SAN is simple but effective. The improvement of SAN is consistent in the reported benchmark.

**Weaknesses:**

The proposed model generalizes existing methods [14, 22], but it reverts to previous approaches when slide number = 1 and an identity statistics prediction model are chosen. While the method performs well empirically, conducting ablation experiments can enhance its validation. For example, testing with a single slice and solely learning statistics prediction would confirm the necessity of slicing.

Moreover, the slicing process requires tuning several important hyperparameters, such as the number of input slices (M) and the number of output slices (K). It would be beneficial if the authors could address the outcome's sensitivity to these hyperparameters.

**Questions:**

The SAN results were slightly worse than the normalization alternatives on the weather dataset. In the paper, the authors attributed this to an over-stationarization issue. Can we detect this problem during training without knowing the generalization performance?

Additionally, does SAN enhance baseline results with a single slice and a learnable statistics prediction module? (See the suggested ablation experiment in the "weakness" section).





**Limitations:**

I recommend conducting ablation studies on slicing and hyperparameter choices (e.g., $M$ and $K$) as mentioned in the "Weakness" and "Questions" sections. Indeed, these are the two new adjustments in the paper compared to earlier work. While these adjustments seem to be useful, the approach's effectiveness will depend on how easily appropriate hyperparameters can be found. The authors could also comment on empirical guidelines for finding new hyperparameters for an unseen task.

---

> ### Author Rebuttal · Authors · 2023-08-09
>
> Thank you for your acknowledgement and valuable feedbacks on our work, we would like to address your concerns as follows.
>
> - Firstly, the suggestion of conducting ablation study on setting slice number to 1 is of great value. Considering both the training efficiency and rebuttal space limitation, we provide additional experiments on such setting with DLinear on 4 datasets in the following table:
>
>   |                 |                 | DLinear_SAN |           | DLinear_1_slice |         | DLinear_wo_SAN |           |
>   | --------------- | --------------- | ----------- | --------- | --------------- | ------- | -------------- | --------- |
>   | **Dataset**     | **Pred_length** | **MSE**     | **MAE**   | **MSE**         | **MAE** | **MSE**        | **MAE**   |
>   | **Electricity** | 96              | **0.137**   | **0.234** | 0.143           | 0.242   | 0.140          | 0.237     |
>   |                 | 192             | **0.151**   | **0.247** | 0.157           | 0.255   | 0.153          | 0.250     |
>   |                 | 336             | **0.166**   | **0.264** | 0.842           | 0.760   | 0.168          | 0.267     |
>   |                 | 720             | **0.201**   | **0.295** | 0.868           | 0.770   | 0.203          | 0.301     |
>   | **Exchange**    | 96              | **0.085**   | 0.214     | **0.085**       | 0.220   | 0.086          | **0.213** |
>   |                 | 192             | 0.177       | 0.317     | 0.163           | 0.309   | **0.161**      | **0.297** |
>   |                 | 336             | **0.294**   | **0.407** | 0.303           | 0.422   | 0.338          | 0.437     |
>   |                 | 720             | **0.726**   | **0.649** | 0.757           | 0.656   | 0.999          | 0.755     |
>   | **Weather**     | 96              | **0.152**   | **0.210** | 0.158           | 0.221   | 0.175          | 0.237     |
>   |                 | 192             | **0.196**   | **0.254** | 0.203           | 0.264   | 0.217          | 0.275     |
>   |                 | 336             | **0.246**   | **0.294** | 0.265           | 0.323   | 0.263          | 0.314     |
>   |                 | 720             | **0.315**   | **0.346** | 0.319           | 0.353   | 0.325          | 0.366     |
>   | **ETTh1**       | 96              | 0.383       | **0.399** | 0.689           | 0.552   | **0.377**      | **0.399** |
>   |                 | 192             | 0.419       | **0.419** | 0.705           | 0.566   | **0.417**      | 0.426     |
>   |                 | 336             | **0.437**   | **0.432** | 0.709           | 0.581   | 0.464          | 0.461     |
>   |                 | 720             | **0.446**   | **0.459** | 0.708           | 0.599   | 0.493          | 0.505     |
>
>   The results demonstrate that though SAN could enhance baseline results with a single slice under certain settings, the multi-slicing approach performs better consistently in the benchmark. This validates our hypothesis that the distribution shift in time series data happens rapidly, therefore a slicing normalization is required to better remove the non-stationarity.
>
> - Secondly, we believe that there is an invariant statistic for a dataset (period for example), therefore we share the slicing length for both input series and output series. Consequently, **the number of input slices (M) and the number of output slices (K) are determined by the slicing length**. In our initial submission, we had already conducted the ablation study on this parameter and the results are provided in the Table 4 in the supplementary materials. Specifically, we utilized SCINet as the backbone under the multi-variate forecasting setting to test the effect of different slicing length on our method. We also attach the results in the following for better illustration.
>
>   | slicing length | 6         |           | 12        |           | 24        |           | 48    |       |
>   | -------------- | --------- | --------- | --------- | --------- | --------- | --------- | ----- | ----- |
>   |                | MSE       | MAE       | MSE       | MAE       | MSE       | MAE       | MSE   | MAE   |
>   | Electricity    | 0.210     | **0.305** | 0.207     | **0.305** | **0.206** | 0.307     | 0.208 | 0.307 |
>   | Exchange       | **0.892** | **0.712** | 0.895     | **0.712** | 0.901     | 0.715     | 0.898 | 0.714 |
>   | Traffic        | 0.612     | **0.376** | 0.608     | **0.373** | **0.607** | 0.381     | 0.611 | 0.382 |
>   | Weather        | **0.338** | 0.366     | **0.338** | **0.365** | 0.340     | 0.367     | 0.339 | 0.366 |
>   | ILI            | **2.487** | **1.063** | 2.680     | 1.118     | n/a       | n/a       | n/a   | n/a   |
>   | ETTh1          | 0.491     | 0.475     | **0.488** | 0.474     | 0.489     | **0.473** | 0.492 | 0.474 |
>   | ETTh2          | 0.440     | 0.465     | **0.435** | 0.460     | 0.437     | **0.459** | 0.443 | 0.462 |
>   | ETTm1          | 0.495     | 0.469     | **0.450** | **0.441** | 0.611     | 0.503     | 0.463 | 0.448 |
>   | ETTm2          | **0.391** | 0.406     | **0.391** | **0.405** | 0.392     | **0.405** | 0.403 | 0.415 |
>
>   Obviously, the results suggest that our proposed SAN method is resilient to changes in slicing length. When it comes to finding new hyperparameters (the slicing length here) for unseen tasks, we recommend using a heuristic approach based on physical properties like cyclical or seasonal patterns. While this method may not yield optimal parameters every time, SAN has shown its ability to achieve near-optimal performance even with sub-optimal slicing lengths.
>
> - Finally, by conducting statistical analysis, such as using the ADF test, we can assess the stationarity of a given time series. Based on these results, it becomes possible to adjust the normalization strength of SAN in order to prevent over-stationarity.
>
> We hope that our responses have adequately addressed your concerns, and we appreciate the opportunity to clarify our work.

---

> > ### Comment · Reviewer_roaz · 2023-08-14
> > **Thank you**
> >
> > Thank you for your response. The two sets of experiments above addressed my questions. I have increased my score.

---

### Decision · Program_Chairs · 2023-09-21

**Decision:**

Accept (poster)

**Comment:**

The paper presents a new method for flexible normalization and denormalization of time series, reducing the non-stationarity within slices of the time series, and predicting the statistics of the target window. The reviewers have a positive outlook on the paper, finding the problem to be well motivated and appreciating the method's generalizability and effectiveness, as well as the quality of the presentation. The reviewers had some concerns, the main ones being possible information loss, sensitivity to hyperparameters and the need for further baselines. During the discussion phase, the authors competently addressed the comments, leading to the reviewers unanimously agreeing that the paper should be accepted, a recommendation which I am also pleased to make.